# Greater than the Sum of Its Parts: Building Substructure into Protein Encoding Models

**Robert Calef**[1,2,*]**, Arthur Liang**[1,*]**, Manolis Kellis**[1]**, Marinka Zitnik**[2]
[1]MIT, [2]Harvard University
{rcalef,artliang,manoli}@mit.edu
marinka@hms.harvard.edu

## Abstract

Protein representation learning has advanced rapidly with the scale-up of sequence and structure supervision, but most models still encode proteins either as per-residue token sequences or as single global embeddings. This overlooks a defining property of protein organization: proteins are built from recurrent, evolutionarily conserved substructures that concentrate biochemical activity and mediate core molecular functions. Although substructures such as domains and functional sites are systematically cataloged, they are rarely used as training signals or representation units in protein models. We introduce Magneton, an environment for developing substructure-aware protein models. Magneton provides (1) a dataset of 530,601 proteins annotated with over 1.7 million substructures spanning 13,075 types, (2) a training framework for incorporating substructures into existing protein models, and (3) a benchmark suite of 13 tasks probing representations at the residue, substructural, and protein levels. Using Magneton, we develop substructure-tuning, a supervised fine-tuning method that distills substructural knowledge into pretrained protein models. Across state-of-the-art sequence- and structure-based models, substructure-tuning improves function prediction, yields more consistent representations of substructure types never observed during tuning, and shows that substructural supervision provides information that is complementary to global structure inputs. The Magneton environment, datasets, and substructure-tuned models are all openly available at https://github.com/rcalef/magneton.

## 1 Introduction

Protein representation learning has progressed from models trained on large sequence databases (Rives et al., 2021; Elnaggar et al., 2022) to models incorporating experimentally determined or predicted structures (Gligorijević et al., 2021; Zhang et al., 2022b), enabling advances in folding (Lin et al., 2023), function prediction (Rao et al., 2019), and variant effect prediction (Meier et al., 2021; Brandes et al., 2023). However, these models have largely ignored the recurrent and modular composition of proteins, which introduces substantial challenges. Protein substructures occur at multiple spatial and functional scales, from local motifs spanning only a handful of residues to domains that cover large fractions of proteins (Durairaj et al., 2023). They are often non-contiguous in sequence space, making them difficult to encode with standard sequence-based protein models (Ovchinnikov & Huang, 2021). A single residue can belong to several overlapping substructures, inducing hierarchical and context-dependent relationships that are not naturally handled by flat representations. Finally, annotated substructures are distributed in a long-tailed fashion, with abundant secondary structure elements but scarce examples of specialized motifs, complicating the design of training objectives and evaluation protocols (Durairaj et al., 2023).

---

*Equal contribution

These challenges arise because proteins are not uniform chains but are organized into recurrent, modular substructures that provide a natural multiscale vocabulary for their representation (Sun et al., 2025). At the finest level are amino acids, which assemble into secondary structure elements such as alpha helices and beta sheets; these in turn combine into higher-order motifs and domains such as beta barrels and zinc fingers (Figure 1A). These substructures are responsible for core molecular functions of proteins, such as coordinating metal ions for reaction catalysis or binding to other proteins as parts of cellular signaling networks, and their importance is underscored by their occurrence in proteins across the tree of life. Decades of biological research has led to the categorization of these recurrent substructures, resulting in large databases that exhaustively annotate these elements across proteins (Sonnhammer et al., 1997; Paysan-Lafosse et al., 2025; Blum et al., 2025a). However, prevailing protein representation learning methods still rely on self-supervised objectives that operate at the scale of single amino acids, such as masked language modeling or structural denoising, or occasionally operate on full proteins (Yu et al., 2023). This is despite abundant evidence that evolutionarily conserved substructures are key components of protein function (Rossman & Liljas, 1974). In this work, we ask, *how should we systematically incorporate decades of biological knowledge about protein substructures into protein encoding models?*

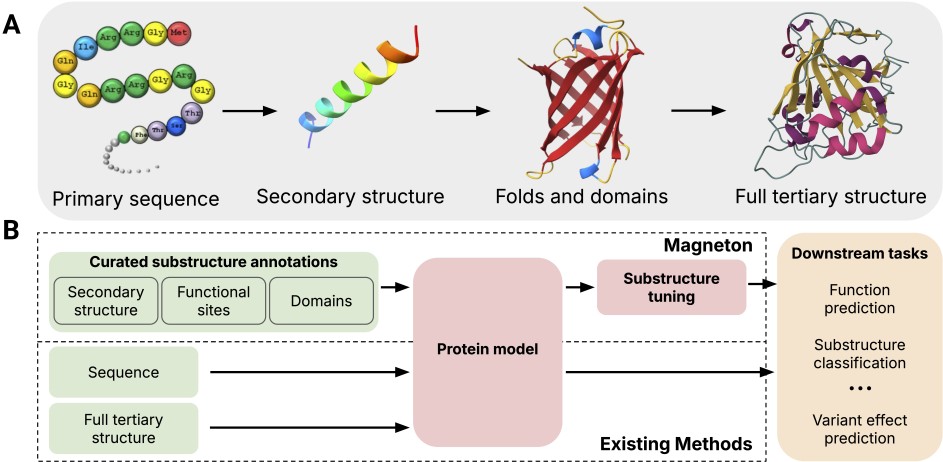

**Figure 1: Overview of protein structure and the Magneton environment.** (A) Proteins are built from modular substructures that assemble into full structures. (B) Magneton leverages decades of substructure research to provide an environment for developing and evaluating substructure-aware models.

While there exists a growing body of work exploring how to best integrate protein sequence and structure into a single model, either via direct incorporation of structural tokens (Su et al., 2023; Li et al., 2024; Hayes et al., 2025; Lu et al., 2025; Yuan et al., 2025) or finetuning of sequence models to better align with structural representations (Zhang et al., 2024b; Ouyang-Zhang et al., 2025), there are few examples of incorporating substructure information into protein encoding models. GearNet (Zhang et al., 2022b) uses a multi-view contrastive objective and cite recurrent substructures as motivation, but use multiple views of subsets of the same protein rather than considering recurrent substructures across proteins. The Functional Community Invariance approach (Wang et al., 2025b) employs secondary structure annotations to guide graph augmentations but ignores higher-order substructures. Other threads of work seek to construct hierarchical representations of proteins, either by connecting residues to their exposed surface areas (Somnath et al., 2022; Zhang et al., 2024c; Mallet et al., 2025), or by progressing from all-atom graphs to residue graphs (Wang et al., 2023), but these works pass over protein substructure as a valuable part of the structural hierarchy.

**Present work.** To close this gap, we create a new environment for developing substructure-aware protein models, which we call *Magneton*. Magneton has three main components: (1) a dataset of proteins with curated substructures; (2) a framework for using these substructures to train or finetune protein encoding models; and (3) a benchmark of evaluation tasks that probe the learned representations at the residue, substructure, and protein levels (Figure 1B). By curating data from Pfam (Paysan-Lafosse et al., 2025), InterPro (Blum et al., 2025b), and DSSP (Hekkelman et al., 2025), we create a dataset of 530,601 proteins with over 1.7 million substructural annotations (37

million when including secondary structure) across six substructure classes with 13,075 distinct substructure types.

Using Magneton, we explore *substructure-tuning*, a supervised fine-tuning strategy that distills substructural information into protein encoders. Concretely, we formulate substructure-tuning as classification of evolutionarily-conserved substructures, where residue-level embeddings produced by a base encoder are pooled to construct substructure representations, which are then used to tune the full model with a cross-entropy loss. This objective is model-agnostic, requiring only residue-level embeddings, and naturally extends to multiple structural scales through a multi-task formulation in which each substructure class is assigned its own prediction head and the total loss is the sum across scales.

Our key contributions are:

- We present Magneton, an environment for exploring substructure-aware protein models, consisting of a large-scale curated dataset of substructural annotations, an associated Python library for training protein encoding models, and a suite of 13 evaluation tasks spanning residue, substructure, protein, and interaction levels.
- We introduce substructure-tuning, a supervised fine-tuning method for distilling substructural information into pretrained models. We exhaustively evaluate its design space across highly local substructures (e.g., active sites spanning 10 residues) to larger domains, and apply it to six state-of-the-art sequence-only and sequence-structure encoders.
- We show that substructure-tuning improves models' ability to represent protein function and yields consistent gains on function-related prediction tasks.
- We find that substructure-tuning produces more consistent representations of substructures of the same type, including types never seen during training, demonstrating that substructure-tuning encourages models to learn general features of functional substructures.
- The above results hold for both sequence-only models and sequence-structure models, showing that substructural information is complementary to global structure.

We envision that Magneton will catalyze the integration of protein substructures into protein models and motivate the development of new approaches and inductive biases that incorporate decades of knowledge about protein structure across scales.

## 2 RELATED WORK

**Integrating structure- and function-based inductive biases into protein sequence models.** A large body of work has explored distilling auxiliary modalities into protein sequence models. Some methods incorporate free-text descriptions, such as Gene Ontology terms (Zhang et al., 2022a) or SwissProt annotations (Xu et al., 2023). The majority, however, focus on structural information. Explicit approaches integrate structure directly, either through structure graphs (Zhang et al., 2023) or structural tokenization (Su et al., 2023; Li et al., 2024). Structural distillation methods instead use structure only at training time, preserving sequence-only inference. For example, Implicit Structure Model (ISM) (Ouyang-Zhang et al., 2025) trains residue-level predictors on tokens from a structural autoencoder, while ESM-S (Zhang et al., 2024b) distills global structural information via fold classification. S-PLM (Wang et al., 2025a) employs contrastive learning to align representations of an ESM encoder with those of a contact-map encoder. Magneton differs by focusing on protein *substructures* rather than only residue-level or global structural signals. It provides large-scale curated annotations of conserved substructures and a framework for supervised finetuning on these elements to encode modular, recurrent units of protein organization. This is orthogonal to existing sequence-structure integration and structural distillation approaches.

**Substructure-aware training and hierarchical models.** Protein substructure admits a hierarchical view, but most hierarchical modeling approaches focus on geometric relations rather than functional substructures. Some methods connect residues to exposed surface areas (Somnath et al., 2022; Zhang et al., 2024c; Mallet et al., 2025), while others connect residues to constituent atoms (Wang et al., 2023). Few approaches incorporate substructural information directly. GearNet (Zhang et al., 2022b) uses a multiview contrastive objective that samples local regions within a protein, but supervision is restricted to intra-protein partitions rather than conserved substructures across proteins. SES-Adapter (Tan et al., 2024) augments sequence models with cross-attention to DSSP-derived secondary structure tokens, but does not extend beyond this single level of annotation. Protein lan-

guage models such as xTrimoPGLM (Chen et al., 2025) use span-masking, but the masked spans are random residue segments rather than biologically-defined substructures. ESM3 (Hayes et al., 2025) introduces multi-track tokenization, including secondary structure and function tracks, where the function track is derived from ontology terms often correlated with substructural annotations. However, the learning remains self-supervised and intra-protein, without supervision on conserved substructures across proteins. Magneton differs by providing annotations of conserved substructures across proteins and by defining supervised training objectives that operate directly on these annotations. This design moves beyond local partitions, random spans, or ontology proxies, enabling systematic study of substructure-aware modeling across residue-, motif-, domain-, and protein-level representations.

**Geometric protein models.** Geometric deep learning has been widely applied to proteins, with models developed for folding (Jumper et al., 2021; Abramson et al., 2024), structure design (Passaro et al., 2025; Watson et al., 2023; Huang et al., 2024), and representation learning (Jing et al., 2020; Fang et al., 2025). These approaches operate at the atom scale (Qu et al., 2025; Widatalla et al., 2025) and encode spatial coordinates of all atoms to model global protein geometry. Magneton addresses a complementary problem: representing recurrent substructures that span residues, motifs, and domains, and recur across proteins. Rather than optimizing directly on atomic coordinates or global geometry, Magneton introduces supervised objectives on conserved substructures, providing functional supervision across structural scales. Substructural objectives can also be integrated with atom-scale geometric encoders to yield models that capture fine-grained geometry and functional modularity.

## 3 METHODS

**Preliminaries.** Two possible views of a protein $P$ are the residue-level, $P = (a_1, \ldots, a_l)$ where $a_i$ is the $i$'th residue in the primary sequence, and the substructure-level, $P = (s_1, \ldots, s_n)$ where each $s_i$ represents a substructure contained within a protein. Other views are possible (*e.g.* atom-level), but these two views are the most relevant for our work. In the substructure view, each substructure is a subset of $k$ residues, $s_i = \{a_j\}_{j=1}^{j=k}$, where the residues $a_j$ may or may not be contiguous in the primary sequence. Since substructures exist at multiple scales, a given residue may be a member of multiple, possibly overlapping, substructures, *e.g.* a residue may be part of a secondary structure element, such as a beta strand, that is itself part of a larger fold, such as a beta barrel. It is also possible for a given residue to not be included in any annotated substructure. While the substructure view of a protein is common in the biological community (Rose, 1979; Vogel et al., 2004; Alberts et al., 2002), there is a lack of curated datasets for exploring it in the context of protein modeling.

### 3.1 MAGNETON ENVIRONMENT

Magneton is an environment for developing substructure-aware protein models, and consists of three main parts: (1) a curated dataset of proteins with annotated substructures, (2) a framework for using this dataset for substructure-aware training, and (3) an integrated benchmark of evaluation tasks that probe a model's learned representations at multiple structural scales.

**Dataset.** We use the `2024_06` release of UniProtKB/TrEMBL (The UniProt Consortium, 2025) as our core protein dataset, containing roughly 254M proteins. We obtain annotations of 8-class secondary structure from DSSP (Kabsch & Sander, 1983; Hekkelman et al., 2025) and annotations of higher-order structures (Homologous superfamilies, domains, conserved sites, active sites, binding sites) from the 103.0 release of InterPro (Blum et al., 2025a). We process these raw releases into Magneton's core datatypes representing a protein and its associated substructures. Due to the scale of the dataset at this stage and the size of protein structure data, we focus our further exploration on the manually curated SwissProt subset of UniProtKB, but make the processed version of the full UniProtKB/TrEMBL dataset available to the community. We obtain amino acid sequences from UniProtKB and predicted structures from AlphaFold DB (Varadi et al., 2022). For consistency across sequence-based and structure-based models, we subset the SwissProt dataset to only proteins with calculated structures in the current (Nov 2022) release of AlphaFold DB, leaving 530,601 proteins. Additional details on the dataset, processing, and example substructures can be found in Appendix A.1.1.

| Substructure class | Unique types (pre-filter) | Total occurrences (pre-filter) | Unique types (post-filter) | Total occurrences (post-filter) | Median protein span |
|---|---|---|---|---|---|
| Homologous superfamily | 2978 | 1.09M | 1133 | 1.05M | 50% (137 AA) |
| Domain | 9133 | 389K | 917 | 301K | 34.8% (127 AA) |
| Conserved site | 739 | 175K | 356 | 162K | 5.18% (16 AA) |
| Binding site | 67 | 20.1K | 48 | 19.0K | 4.28% (16 AA) |
| Active site | 132 | 31.1K | 82 | 29.2K | 3.47% (12 AA) |
| Secondary structure | 8 | 35.2M | 8 | 35.2M | 0.94% (3.4 AA) |
| Total w/o secondary structure | 13075 | 1.71M | 2542 | 1.56M | — |

Table 1: **Summary of Magneton substructure dataset (SwissProt subset).** Before and after refer to filtering out rare substructures. Median protein span is the median length of a type of substructure, expressed as a percentage of the protein and as an absolute amino acid count.

To focus learning efforts on substructures where sufficient data is present, we create a restricted label set of substructures that occur at least 75 times in the SwissProt dataset, corresponding to retaining only the top 10% most frequently occurring domains. While this may seem stringent, this retains the vast majority of actual substructure occurrences across types, since many substructures have very few occurrences. We additionally generate versions of our dataset using more permissive cutoffs (minimum counts of 25 or 10) (Appendix A.1.3). Our published datasets retain all substructure annotations to enable future research by the community. Table 1 summarizes the different classes of substructures, their counts, number of types, and typical span on the protein. As expected for substructural elements, the majority of the substructures span less than 10% of the annotated protein, with the scale varying by the class of substructure. We then split this dataset into training, validation, and test sets using the AFDB50 sequence-based clusters (Barrio-Hernandez et al., 2023), ensuring that sequences sharing more than 50% identity and 90% overlap are assigned to the same split.

**Evaluation benchmark.** To provide a holistic evaluation of substructure-focused protein modeling within Magneton, we integrate numerous evaluation tasks from the community. These tasks probe a model's learned representations at multiple scales: individual residues, substructures, proteins, and protein interactions (Table 2). At the residue-level, we include contact prediction (Rao et al., 2019), zero-shot prediction of variant effects (Notin et al., 2023), and multiple types of functional residue prediction tasks (Dallago et al., 2021; Yuan et al., 2025); at the substructure-level, we include multiclass substructure classification problems derived from the Magneton dataset itself; at the protein-level, we include function prediction (GO and EC terms) (Gligorijević et al., 2021), subcellular localization (Almagro Armenteros et al., 2017), and fitness prediction (Rao et al., 2019). Finally, we include a human PPI prediction task (Pan et al., 2010; Xu et al., 2022). Full details of evaluation datasets can be found in Appendix A.1.5.

## 3.2 SUBSTRUCTURE REPRESENTATION AND TUNING

Given the dataset in Magneton, we now have a large collection of proteins $\mathcal{P}$, where each protein has curated substructural annotations, $P = (s_1, \ldots, s_k); P \in \mathcal{P}$. We first use this dataset to assess whether existing protein models can generate meaningful representations of substructures. Specifically, for a protein model $f$, we construct a representation of each substructure $s_j \in P$ by calculating residue-level embeddings, $f(P) = (v_1, \ldots, v_l), v_l \in \mathbb{R}^d$ where $v_i$ is the embedding of residue $a_i$. We then perform a substructure pooling operation over the constituent residues of $s$, $f(s) = \texttt{pool}(\{v_i : a_i \in s\}, f(s) \in \mathbb{R}^d$, where $\texttt{pool}$ can be any arbitrary pooling operation. These substructure-level representations are then input to a classifier over the possible substructure labels for the final substructure classification task. Since a substructure's constituent residues are given to the model, this is a *diagnostic task* meant to probe each model's ability to represent sub-

| Scale | Task | Task type | Metric | Data source |
|---|---|---|---|---|
| Interaction | Human PPI prediction | Binary | Accuracy | Pan et al. |
| Protein | Gene Ontology prediction | Multilabel | $F_{\max}$ | Gligorijević et al. |
| | Enzyme Commission prediction | Multilabel | $F_{\max}$ | |
| | Subcellular localization | Multiclass | Accuracy | Almagro Armenteros et al. |
| | Binary localization | Binary | Accuracy | |
| | Thermostability prediction | Regression | Spearman's $\rho$ | Rao et al. |
| Substructure | Substructure classification | Multiclass | Macro accuracy | Ours |
| Residue | Contact prediction | Binary | Precision@L | Rao et al. |
| | Variant effect prediction | Regression | Spearman's $\rho$ | Notin et al. |
| | Binding residue categorization | Multilabel | $F_{\max}$ | Dallago et al. |
| | Functional site prediction | Binary | AUROC | Yuan et al. |

**Table 2: Evaluation tasks contained within Magneton.** Grouped by the scale of structural representation they interrogate.

structures, not a task meant to measure the ability to identify previously unannotated substructures. For this diagnostic assessment, we freeze the parameters of the underlying protein model and train only the substructure classification head.

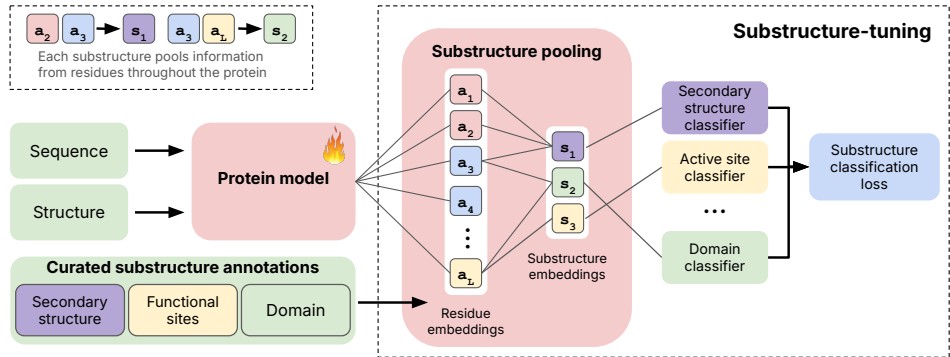

**Figure 2: Overview of using Magneton for substructure-tuning.** Given a pre-trained protein model, substructure-tuning first pools residue-level embeddings to create substructure representations, which are then used for supervised finetuning via substructure type-specific classifier heads.

We next explore imbuing existing protein models with substructural information. In a process we refer to as *substructure-tuning*, we again perform the substructure classification task outlined above, but with finetuning of the original protein model's parameters (Figure 2) to encourage the model to distinguish between the many different types of biologically-relevant substructures in our dataset. Although we use supervised finetuning, other losses, such as a contrastive objective (van den Oord et al., 2019), could also be used. The substructure-tuning process is compatible with any finetuning method, including parameter-efficient methods such as LoRA (Hu et al., 2021) for larger base models. We perform substructure-tuning using the Magneton training set and explore tuning with different substructure types as well as their combinations. When finetuning with multiple substructure classes, each class uses its own predictor module with the cross-entropy loss across all types summed to form the final substructure classification loss.

### 3.3 IMPLEMENTATION DETAILS

For our experiments, we select base protein models that represent state-of-the-art models across a range of model sizes and modality inputs. For sequence-based models, we use ESM2-150M and -650M (Lin et al., 2023) and ESM-C 300M and 600M (ESM Team, 2024). For models that incorporate protein structure, we use SaProt (Su et al., 2023) and ProSST-2048 (Li et al., 2024), both of which use both protein sequence and structure. We opt to exclude purely structural models as their performance is generally below that of the sequence-structure models we've included.

For substructure classification and tuning, we use single hidden layer MLPs, where the hidden dimension size matches that of the base model, as our prediction modules, and mean pooling for the substructure pooling operation (see Appendix A.4.1 for exploration of alternate pooling methods). For substructure-tuning, we perform full finetuning of the base model. We use elastic weight consolidation (EWC) (Kirkpatrick et al., 2017) to avoid catastrophic forgetting of the base model's original objective. Detailed training methodology is available in Appendix A.2.1.

For supervised downstream evaluations, we train head models on top of either the original base model or the substructure-tuned base model. For these evaluations, we freeze the base model to focus on evaluating the representations learned during substructure-tuning. Results across all tasks and models were generated within the Magneton environment and use identical datasets and splits. We unfortunately exclude ProSST from the functional site prediction and contact prediction tasks due to its incompatibility with experimental structures from PDB. Full training details for all models and evaluation tasks are available in Appendix A.2.2.

# 4 EXPERIMENTS

## 4.1 SUBSTRUCTURE REPRESENTATION ASSESSMENT

Table 3 shows that base models are readily able to produce effective representations of substructures across scales, with structure-based models generally outperforming sequence-only models. We also find that models can correctly classify substructures within proteins that contain multiple substructures (*e.g.*, accurately classifying all domains within a single protein containing multiple domains), indicating that classification relies on local structural cues rather than global structural similarity (Figure 3A). While performance degrades for some rarer substructures, we generally see high accuracy even for rare substructures (Figure 3B).

| Model | Homologous superfamily | Domain | Conserved site | Binding site | Active site | Secondary structure |
|---|---|---|---|---|---|---|
| ESM2-150M | 0.899 | 0.969 | 0.988 | 1.000 | 0.995 | 0.827 |
| +ST | 0.925 | 0.983 | 0.991 | 0.999 | 0.994 | 0.916 |
| ESM2-650M | 0.926 | 0.982 | 0.986 | 1.000 | 0.995 | 0.892 |
| +ST | 0.902 | 0.967 | 0.986 | 1.000 | 0.996 | 0.938 |
| ESM-C 300M | 0.913 | 0.962 | 0.990 | 0.998 | 0.994 | 0.863 |
| +ST | 0.946 | 0.982 | 0.983 | 0.999 | 0.996 | 0.757 |
| ESM-C 600M | 0.919 | 0.975 | 0.992 | 0.977 | 0.994 | 0.891 |
| +ST | 0.907 | 0.966 | 0.993 | 0.997 | 0.996 | 0.927 |
| SaProt (650M) | 0.916 | 0.967 | 0.992 | 0.999 | 0.996 | 0.955 |
| +ST | 0.925 | 0.980 | 0.993 | 0.999 | 0.996 | 0.972 |
| ProSST-2048 | 0.888 | 0.945 | 0.995 | 0.996 | 0.993 | 0.927 |
| +ST | 0.879 | 0.976 | 0.991 | 0.991 | 0.995 | 0.961 |

**Table 3: Comparison of substructure classification performance.** Performance on the *diagnostic task* of classifying substructures given their annotated residues, for base and substructure-tuned (+ST) models. All values are macro-averaged accuracy.

## 4.2 SUBSTRUCTURE-TUNING

**Substructure-tuning configurations.** Table 4 shows the results of substructure-tuning with a range of different substructure classes, both individually and their combinations, as measured by downstream evaluation tasks. Due to the large number of possible configurations, we restricted this initial exploration to a single model (ESM-C 300M), a subset of evaluation tasks, and a selection of the $2^6$ possible substructure class combinations aimed at exploring combinations of substructure classes across scales.

Our exploration of substructure configurations revealed the following: 1) The effects of substructure-tuning are largely consistent across the selected substructure types used, with performance boosts in

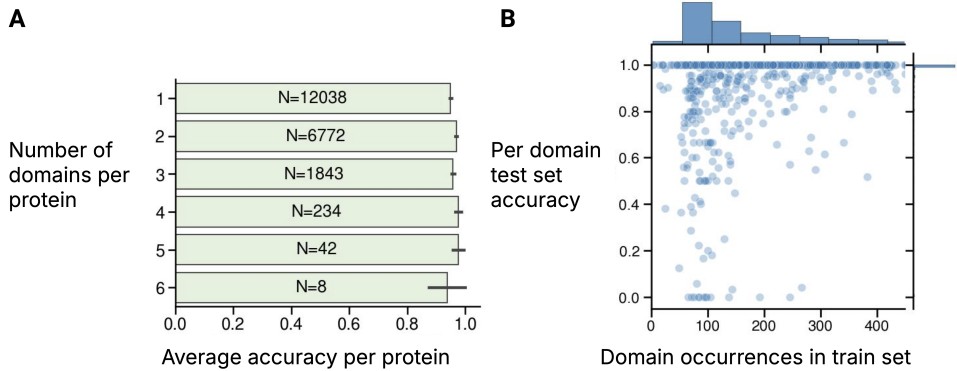

**Figure 3:** **(A) Domain classification uses local cues.** Even within proteins containing multiple domains, classification accuracy remains high for all contained domains. Labels within bars show the number of test set proteins containing that number of domains. **(B) Domain classification accuracy as a function of training set representation.** Results shown for ESM-C 300M.

| Substructures used | EC | GO:BP | GO:CC | GO:MF | Localization (Accuracy) | | Thermostability (Spearman's $\rho$) | Zero-shot DMS ( Spearman's $\rho$) |
|---|---|---|---|---|---|---|---|---|
| | | | $F_{\max}$ | | Binary | Subcellular | | |
| None | 0.688 | 0.307 | **0.416** | 0.429 | 0.871 | 0.703 | 0.648 | **0.432** |
| H_____ | 0.805 | 0.312 | 0.395 | 0.518 | 0.851 | 0.632 | 0.662 | 0.308 |
| _D____ | 0.776 | 0.307 | 0.403 | 0.501 | 0.811 | 0.640 | **0.666** | 0.340 |
| __C___ | 0.749 | 0.318 | 0.398 | 0.491 | 0.870 | **0.706** | 0.661 | 0.402 |
| ___B__ | 0.745 | 0.315 | 0.415 | 0.478 | 0.852 | 0.686 | 0.663 | 0.423 |
| ____A_ | 0.794 | 0.318 | 0.403 | 0.518 | 0.851 | 0.639 | 0.663 | 0.340 |
| _____S | 0.618 | 0.297 | 0.379 | 0.381 | 0.823 | 0.587 | 0.612 | 0.264 |
| HD____ | 0.774 | 0.316 | 0.388 | 0.500 | 0.847 | 0.606 | 0.639 | 0.302 |
| H____S | 0.765 | 0.297 | 0.395 | 0.466 | **0.883** | 0.651 | 0.644 | 0.346 |
| HD___S | 0.754 | 0.318 | 0.413 | 0.473 | 0.868 | 0.633 | 0.658 | 0.350 |
| H_CBA_ | 0.800 | 0.322 | 0.389 | 0.515 | 0.857 | 0.611 | 0.663 | 0.340 |
| _D___S | 0.751 | 0.308 | 0.384 | 0.462 | 0.872 | 0.646 | 0.643 | 0.369 |
| _DCBA_ | **0.815** | **0.329** | 0.395 | **0.525** | 0.851 | 0.662 | 0.659 | 0.369 |
| __CBA_ | 0.761 | 0.325 | 0.403 | 0.488 | 0.879 | 0.681 | 0.660 | 0.410 |
| ___BA_ | 0.740 | 0.319 | 0.406 | 0.467 | 0.841 | 0.677 | 0.656 | 0.418 |
| __CBAS | 0.719 | 0.313 | 0.393 | 0.453 | 0.839 | 0.666 | 0.636 | 0.379 |
| HDCBAS | 0.760 | 0.315 | 0.383 | 0.457 | 0.832 | 0.624 | 0.640 | 0.359 |

**Table 4: Comparison of substructure-tuning configurations.** Performance across tasks for ESM-C 300M with a range of substructure-tuning configurations. For each configuration, the substructures used are indicated by the presence of that substructure type's single-letter code: H=Homologous superfamily, D=Domain, C=Conserved site, B=Binding site, A=Active site, S=Secondary structure; an underscore (_) means that substructure type was not used.

tasks related to protein function (GO:MF, GO:BP, EC, Thermostability) and neutral to negative effects on localization tasks (GO:CC, Binary localization, Subcellular localization) and residue-level variant-effect prediction. 2) These effects are present even when tuning with very small substructures, such as active sites, which typically consist of only 12 amino acids (median protein span of 3.47%). Based on these results, we selected the combination of active site, binding site, and conserved site as the substructure-tuning configuration for use in the full set of models and benchmarks, as this configuration represented a balance of positive gains in function-related tasks and neutral effects in localization and variant-effect tasks at the residue level.

**Substructure-tuning across models.** Tables 5 and 6 show how the selected substructure-tuning configuration affects the downstream performance of the full set of base protein models across protein-level and residue-level tasks, respectively (see Appendix Tables A.9 and A.10 for uncertainty estimates). The full evaluation across models and benchmarks led to the following conclusions: 1) Results across models are consistent with the initial exploration: performance boosts in function-related tasks and neutral to negative effects on localization and residue-level tasks. 2) Importantly, these results hold for models that already incorporate protein structure as an input (ProSST-2048 and SaProt), suggesting complementarity between structural and substructural information.

| Model | EC | GO:BP | GO:CC | GO:MF | Localization (Accuracy) | | Thermostability (Spearman's $\rho$) | Human PPI (AUROC) |
|---|---|---|---|---|---|---|---|---|
| | | $F_{\max}$ | | | Binary | Subcellular | | |
| ESM2-150M | 0.727 | 0.316 | 0.416 | 0.441 | 0.869 | 0.694 | 0.627 | 0.933 |
| +ST | 0.742 | 0.324 | 0.415 | 0.473 | 0.866 | 0.679 | 0.582 | 0.919 |
| ESM2-650M | 0.755 | 0.319 | 0.431 | 0.486 | 0.876 | 0.710 | 0.643 | 0.939 |
| +ST | 0.745 | 0.321 | 0.440 | 0.534 | 0.895 | 0.749 | 0.655 | 0.935 |
| ESM-C 300M | 0.688 | 0.307 | 0.416 | 0.429 | 0.871 | 0.703 | 0.648 | 0.917 |
| +ST | 0.761 | 0.325 | 0.403 | 0.488 | 0.879 | 0.681 | 0.660 | 0.933 |
| ESM-C 600M | 0.701 | 0.312 | 0.403 | 0.436 | 0.863 | 0.713 | 0.668 | 0.927 |
| +ST | 0.780 | 0.319 | 0.385 | 0.527 | 0.872 | 0.635 | 0.667 | 0.902 |
| SaProt (650M) | 0.778 | 0.326 | 0.453 | 0.538 | 0.887 | 0.784 | 0.692 | 0.952 |
| +ST | 0.839 | 0.339 | 0.446 | 0.584 | 0.896 | 0.741 | 0.697 | 0.932 |
| ProSST-2048 | 0.778 | 0.317 | 0.426 | 0.522 | 0.878 | 0.693 | 0.686 | 0.925 |
| +ST | 0.791 | 0.314 | 0.420 | 0.567 | 0.853 | 0.683 | 0.648 | 0.883 |

Table 5: Protein-level task performance for base models and models with substructure-tuning (+ST).

| Model | Binding residue ($F_{\max}$) | Functional site prediction | | Contact Prediction | | | Variant Effect (Spearman's $\rho$) |
|---|---|---|---|---|---|---|---|
| | | Binding | Catalytic | Short | Medium | Long | |
| | | (AUROC) | | (Precision@L) | | | |
| ESM2-150M | 0.379 | 0.871 | 0.910 | 0.487 | 0.452 | 0.289 | 0.342 |
| +ST | 0.327 | 0.852 | 0.890 | 0.460 | 0.445 | 0.285 | 0.262 |
| ESM2-650M | 0.366 | 0.849 | 0.912 | 0.551 | 0.528 | 0.372 | 0.359 |
| +ST | 0.362 | 0.851 | 0.927 | 0.532 | 0.518 | 0.367 | 0.317 |
| ESM-C 300M | 0.367 | 0.851 | 0.923 | 0.339 | 0.364 | 0.174 | 0.432 |
| +ST | 0.411 | 0.866 | 0.910 | 0.350 | 0.374 | 0.180 | 0.410 |
| ESM-C 600M | 0.357 | 0.850 | 0.921 | 0.329 | 0.362 | 0.161 | 0.434 |
| +ST | 0.368 | 0.852 | 0.906 | 0.313 | 0.315 | 0.141 | 0.381 |
| SaProt (650M) | 0.423 | 0.891 | 0.923 | 0.788 | 0.747 | 0.697 | 0.457 |
| +ST | 0.400 | 0.871 | 0.924 | 0.765 | 0.726 | 0.647 | 0.405 |
| ProSST-2048 | 0.375 | N/A | N/A | N/A | N/A | N/A | 0.507 |
| +ST | 0.342 | N/A | N/A | N/A | N/A | N/A | 0.356 |

Table 6: Residue-level task performance for base models and models with substructure-tuning (+ST).

Due to the close relationship between substructures and protein function, we additionally verify that performance increases from substructure-tuning are not trivially attributed to leakage between the Magneton substructure training set and the test sets of the evaluation tasks (Appendix A.1.4). An ablation of EWC finds that it moderates the performance improvements of substructure-tuning, while reducing the amount of degradation in tasks where substructure-tuning has negative effects (Appendix A.2.1). Additionally, we found that substructure-tuning compares favorably to existing methods that distill global structural information into sequence-only models (Appendix A.4.2).

We explored how substructure-tuning interacts with task-specific finetuning by repeating the evaluations above for a subset of models and tasks with full finetuning of the protein model for each task (Appendix A.2.3). We found that task-specific finetuning results in similar performance across models trained with and without substructure-tuning, indicating that aggressive task-specific finetuning may dominate the substructural information imbued during the substructure-tuning process.

**Mechanistic exploration of substructure-tuning.** We next investigated the effects of substructure-tuning on the learned embeddings of the underlying protein models. We found that substructure-tuning greatly increased a model's ability to group substructures of the same type, as measured by silhouette score (Table 7, Appendix Figure A.13). Furthermore, by restricting our analysis to the rare substructure types that were excluded from the Magneton training set entirely, we find that substructure-tuning results in more consistent representations of even substructure types that were never seen during training. This indicates that substructure-tuning encourages models to learn general features of functional substructures, rather than just signatures of specific substruc-

| Model | Homologous superfamily | | Domain | | Conserved site | | Binding site | | Active site | |
|---|---|---|---|---|---|---|---|---|---|---|
| | Seen | Unseen | Seen | Unseen | Seen | Unseen | Seen | Unseen | Seen | Unseen |
| ESM-C 300M | -0.183 | 0.180 | -0.184 | 0.201 | 0.279 | 0.466 | 0.378 | 0.641 | 0.490 | 0.476 |
| +ST | 0.339 | 0.584 | 0.486 | 0.652 | 0.830 | 0.747 | 0.882 | 0.894 | 0.933 | 0.816 |
| SaProt (650M) | 0.079 | 0.301 | 0.122 | 0.412 | 0.534 | 0.623 | 0.613 | 0.796 | 0.714 | 0.701 |
| +ST | 0.478 | 0.684 | 0.554 | 0.717 | 0.796 | 0.764 | 0.843 | 0.938 | 0.912 | 0.866 |

**Table 7: Silhouette scores for substructure types included ("seen") and excluded ("unseen") from training.** Higher silhouette scores indicate tighter clustering of substructures within a type. "Seen" scores are generated using the Magneton test set proteins (*i.e.* "seen" refers to substructure types, not individual proteins).

ture types. These experiments focused on ESM-C 300M and SaProt as representative sequence-only and sequence-structure models.

To understand the task-specific effects of substructure-tuning, we performed a gradient conflict analysis, in which we compared the gradient updates for ESM-C 300M for the substructure classification task and for a set of evaluation tasks, including protein-level function prediction and residue-level classification tasks (Appendix A.3.2). Looking across batches within a task, we found that gradients for the evaluation tasks were highly consistent and gradients for substructure classification had lower, although still positive, similarity. However, substructure classification gradients were close to orthogonal to gradients for the evaluation tasks. While these results do not fully explain the task-specific effects of substructure-tuning, they suggest that the behavior is not due to a simple misalignment between the substructure objective and certain downstream tasks. Instead, we hypothesize that our current instantiation of substructure-tuning biases the model against fine-grained residue-level distinctions, because it explicitly encourages residues within the same substructure to share similar representations.

Finally, we performed an explainability analysis to understand if substructure-tuning increases a model's utilization of substructural information. For the subset of GO:MF terms that can be mapped to domain annotations, we found that substructure-tuning resulted in increased attribution of predictions to residues within domains by an average of 17% over the untuned base model (Appendix A.3.3).

## 5 CONCLUSION AND FUTURE WORK

Our study has several limitations and directions for future work. We focused on a model-agnostic substructure-tuning objective applied to existing protein encoders. While this approach consistently improves function-centric tasks, it yields mixed effects across the full benchmark and can be attenuated by aggressive task-specific finetuning. These findings motivate exploring alternative ways to incorporate substructural information, including architectures and objectives that explicitly integrate substructures. Modifications at the architectural level, such as hierarchical or graph-based encoders, or training objectives that operate across multiple scales simultaneously, may provide a more stable integration strategy. Our current exploration of substructure-tuning focused on which substructure types to use for tuning, but their representation varies greatly across the dataset (e.g., millions of secondary structures, tens of thousands of active sites). Exploring how to best balance or weight these different types is another avenue of future exploration. Finally, our experiments were restricted to SwissProt proteins. Extending to the full UniProtKB and incorporating the long tail of infrequent substructures could enable deeper insights into poorly characterized aspects of protein modularity.

In this work, we have presented the open problem: *how to best incorporate decades of research on protein substructures into protein models?* To this end, we introduced Magneton, an integrated environment for developing and evaluating substructure-aware protein models. Using Magneton, we explored both how well existing models can represent protein substructures and whether a supervised fine-tuning paradigm can be used to effectively imbue those models with substructural information. We found that while this direct, intuitive substructure-tuning approach has both positive and negative effects on downstream tasks, it also encourages models to learn general features of functional substructures and suggests that substructural information is complementary to global structure. Our work lays the foundation for the development of substructure-aware protein models.

## 6 ETHICS STATEMENT

This work involves the analysis of publicly available protein sequence and structure data from established databases (UniProtKB/SwissProt, AlphaFold DB, InterPro, and Pfam). All data used in this study are derived from previously published sources and do not involve human subjects, animal experiments, or the generation of new biological data that require ethical oversight. Our work improves computational methods for understanding protein function, which could contribute to advances in drug discovery and biotechnology. We encourage responsible use of our methods and datasets, which are publicly available, to promote scientific reproducibility and advancement.

## 7 REPRODUCIBILITY STATEMENT

To ensure the reproducibility of our work, we provide the following:

1. All code for Magneton, including data processing pipelines, model training scripts, and evaluation benchmarks, is available at `https://github.com/rcalef/magneton`. The processed datasets are publicly available at `https://huggingface.co/datasets/rcalef/magneton-data`.

2. We provide comprehensive implementation details, including model architectures and hyperparameters, training procedures, optimization details, and data splitting procedures (Methods 3.3).

3. We specify all experimental details, including dataset statistics and preprocessing steps such as substructure filtering criteria and thresholds (Table 1, Appendix A.1.1), as well as evaluation metrics and protocols for all benchmark tasks (Table 2).

4. All experiments can be reproduced using 1-4 NVIDIA A100 GPUs.

The modular design of Magneton facilitates easy plug-and-play usability of our benchmark suite, supporting not only reproducibility but also future research in this area.

## 8 ACKNOWLEDGEMENTS

We gratefully acknowledge the support of NIH R01-HD108794, NSF CAREER 2339524, U.S. DoD FA8702-15-D-0001, ARPA-H Biomedical Data Fabric (BDF) Toolbox Program, Harvard Data Science Initiative, Amazon Faculty Research, Google Research Scholar Program, AstraZeneca Research, Roche Alliance with Distinguished Scientists (ROADS) Program, Sanofi iDEA-iTECH Award, GlaxoSmithKline Award, Boehringer Ingelheim Award, Merck Award, Optum AI Research Collaboration Award, Pfizer Research, Gates Foundation (INV-079038), Aligning Science Across Parkinson's Initiative (ASAP), Chan Zuckerberg Initiative, John and Virginia Kaneb Fellowship at Harvard Medical School, Biswas Computational Biology Initiative in partnership with the Milken Institute, Harvard Medical School Dean's Innovation Fund for the Use of Artificial Intelligence, and the Kempner Institute for the Study of Natural and Artificial Intelligence at Harvard University. RC was supported by the National Science Foundation Graduate Research Fellowship under Grant No. 2141064. Any opinions, findings, conclusions or recommendations expressed in this material are those of the authors and do not necessarily reflect the views of the funders.

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

# A    APPENDIX

## A.1    DATASETS

### A.1.1    MAGNETON SUBSTRUCTURE DATASET

Here we provide additional details on the curated protein substructure dataset that makes up a core part of the Magneton environment.

**Dataset processing.** Below we outline the steps to create the protein substructure dataset:

- We start from the full XML file of all InterPro annotations (`match_complete.xml.gz`) for the 103.0 release, downloaded from the InterPro FTP server [1].

- We parse each XML entry, which each correspond to a single protein, by extracting all `<match>` elements that contain at least one `<ipr>` element. The presence of the `<ipr>` element indicates that the annotation is integrated into InterPro and has been assigned a unique InterPro accession ID. No additional filtering is performed at this stage.

- At this point, the dataset covers all of the approximately 254 million proteins in the `2024_06` release of UniProtKB/TrEMBL. To subset to the SwissProt set, we obtain the set of all SwissProt proteins (UniProt IDs and amino acid sequences) from the `uniprot_sprot-only2024_06.tar.gz` file on the UniProt FTP server [2].

- To ensure consistency between models trained using only sequence data and models trained using sequence and structure data, we further subset the SwissProt set to only proteins contained within the v4 release of AlphaFold DB (`swissprot_cif_v4.tar`)[3].

- We use the CIF files from AlphaFold DB to source secondary structure annotations as calculated using DSSP.

- To select for a non-redundant set of substructural annotations, we select only annotations marked as "representative" within InterPro, when such annotations are available (as of the 103.0 release, these were only available for the "Repeat", "Family", and "Domain" types) [4].

The full set of scripts used for the above steps alongside a detailed `README` file are available in our GitHub repository in the `scripts/dataset_processing` directory.

**Substructure type descriptions.** Here we provide additional information about the various types of substructures contained within the Magneton dataset.

- **Homologous superfamily** - a group of proteins that share a common evolutionary origin, reflected by similarity in their structure, even if sequence similarity is low. Examples include the alpha-helical portion of some viral capsid proteins (IPR008935) and a group of single-stranded DNA-binding transcriptional regulator proteins (IPR009044).

- **Domain** - distinct functional, structural, or sequence units that may exist in a variety of biological contexts. Examples include zinc finger binding domains which serve as binding sites for various types of ligands (one example type is IPR000058) and various types of phosphatase domains which enable regulation of protein phosphorylation (such as IPR000242).

- **Conserved site** - a short sequence that contains one or more conserved residues. Examples include the helix-turn-helix motif found in all known DNA binding proteins that regulate gene expression (IPR000047) and the helix-hairpin-helix motif found in proteins that exhibit non-specific DNA binding activity (IPR000445).

- **Binding site** - a short sequence that contains one or more conserved residues, which form a protein interaction site. Examples include sites for binding copper in various enzymes (IPR001505) and sites for binding proteins with other well-characterized motifs (for example, the IQ motif which binds the EF-hand domain, IPR000048).

---

[1]https://ftp.ebi.ac.uk/pub/databases/interpro/releases/103.0/

[2]https://ftp.uniprot.org/pub/databases/uniprot/previous_releases/release-2024_06/knowledgebase/

[3]https://ftp.ebi.ac.uk/pub/databases/alphafold/v4/

[4]https://interpro-documentation.readthedocs.io/en/latest/represent_dom.html

- **Active site** - a short sequence that contains one or more conserved residues, which allow the protein to bind a ligand. Examples include active sites for catalyzing hydrolysis of DNA and RNA (IPR002071) and active sites for catalyzing the breakdown of lipids (IPR008265).

- **Secondary structure** - conserved local spatial arrangements of a span of amino acids in a protein. Canonical examples are alpha helices and beta sheets. For our work, we use 8-class secondary structure definitions from DSSP [5].

**Example dataset entry.** For illustrative purposes, here we provide an abbreviated example of a single entry in the Magneton dataset in JSONL format:

```
{
  "uniprot_id": "A0A009IHW8",
  "name": "ABTIR_ACIB9",
  "length": 269,
  "entries": [
    {
      "id": "IPR035897",
      "element_type": "Homologous_superfamily",
      "match_id": "G3DSA:3.40.50.10140",
      "element_name": "Toll/interleukin-1 receptor homology (TIR) domain superfamily",
      "representative": false,
      "positions": [
        [
          80,
          266
        ]
      ]
    },
    {
      "id": "IPR000157",
      "element_type": "Domain",
      "match_id": "PF13676",
      "element_name": "Toll/interleukin-1 receptor homology (TIR) domain",
      "representative": false,
      "positions": [
        [
          138,
          231
        ]
      ]
    },
    {
      "id": "IPR000157",
      "element_type": "Domain",
      "match_id": "PS50104",
      "element_name": "Toll/interleukin-1 receptor homology (TIR) domain",
      "representative": true,
      "positions": [
        [
          133,
          266
        ]
      ]
    },
    ...
  ],
  "secondary_structs": [
    {
      "dssp_type": "Alphahelix",
      "start": 3,
      "end": 21
    },
    {
      "dssp_type": "Turn",
      "start": 21,
      "end": 22
    },
    {
      "dssp_type": "Turn",
      "start": 24,
      "end": 26
    },
    ...
  ]
}
```

---

[5]https://pdb-redo.eu/dssp/about

### A.1.2 SUBSTRUCTURE COMPOSITION ANALYSIS

Here we provide exploratory plots to give a sense of the overall size and composition of the different types of substructures contained within the Magneton dataset. These plots show data collected from the SwissProt dataset prior to filtering.

As shown in Table 1, we find that the typical length of a substructure varies widely, with domains spanning hundreds of residues and various functional sites spanning tens of residues (Appendix Figures A.1, A.2). We also find some binding site outliers in terms of length, which may be indicative of annotation artifacts. Despite the wide variance in length for substructures, we find that the amount of the protein they cover is relatively consistent for domains and functional sites (Appendix Figure A.4). We additionally inspect the amino acid composition of the various substructure types, both at the individual amino acid level (Appendix Figures A.5, A.5), and with amino acids grouped by chemical characteristics of their sidechains (Appendix Figure A.7).

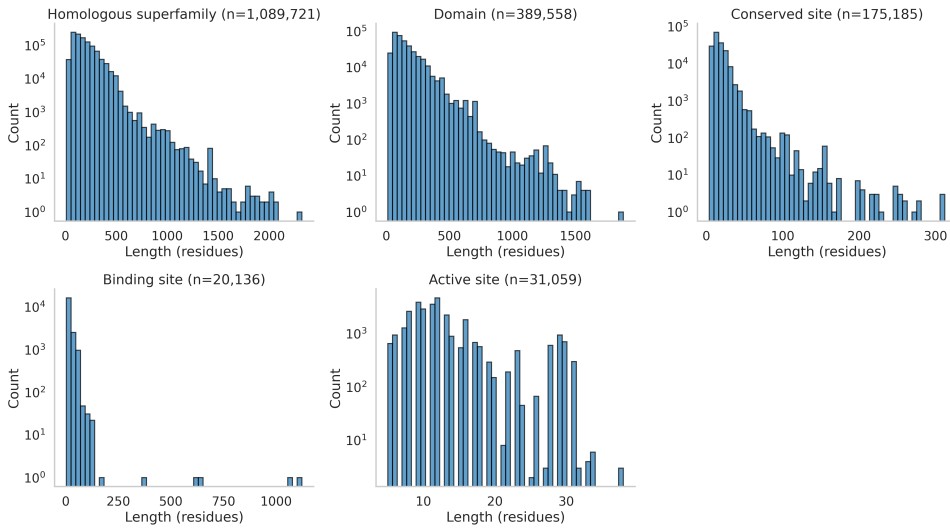

**Figure A.1:** Distribution of substructure lengths by type. Here, length is defined as the total number of residues contained within the substructure, regardless of whether they are contiguous within the sequence.

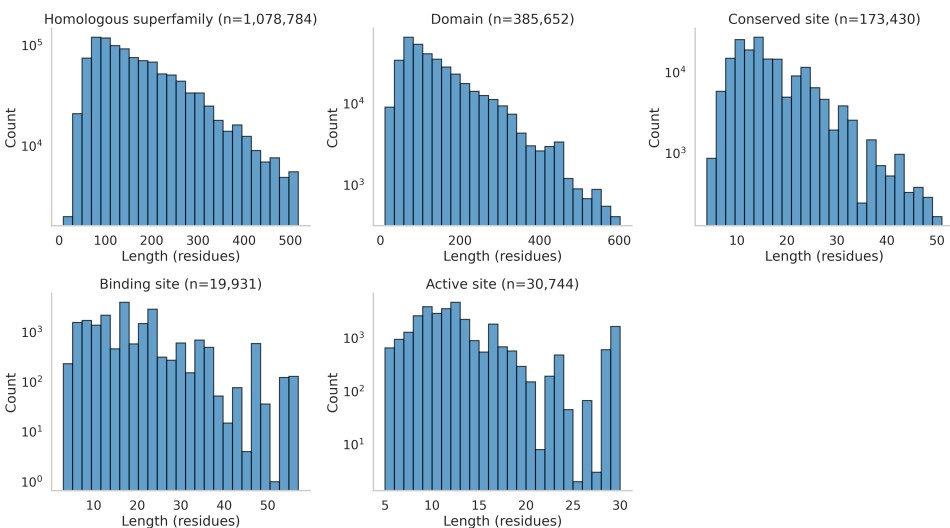

**Figure A.2:** Distribution of substructure lengths by type. Same as Figure A.1, but filtered to remove outliers (greater than 99th percentile of length within that type).

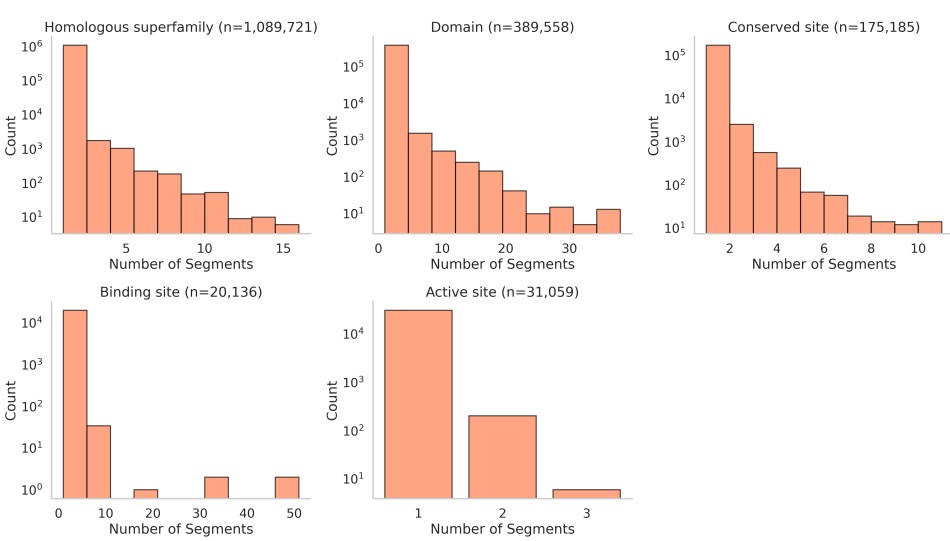

**Figure A.3:** Distribution of number of contiguous segments per substructure, by type.

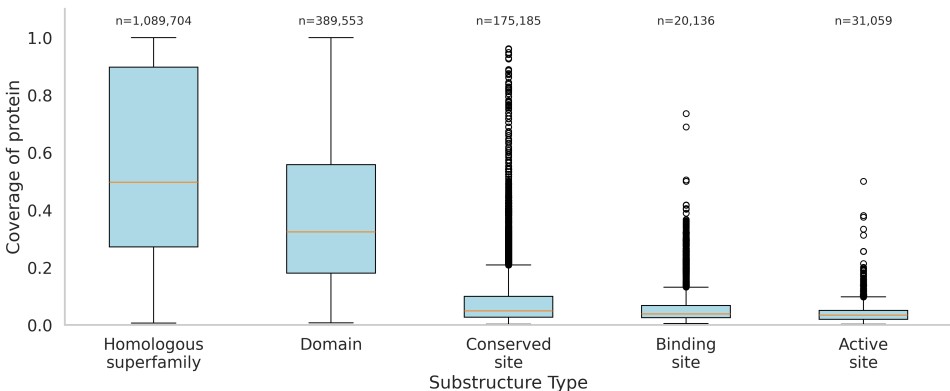

**Figure A.4:** Distribution of protein coverage per substructure. Protein coverage is defined as the total number of residues contained within the substructure divided by the length of the protein.

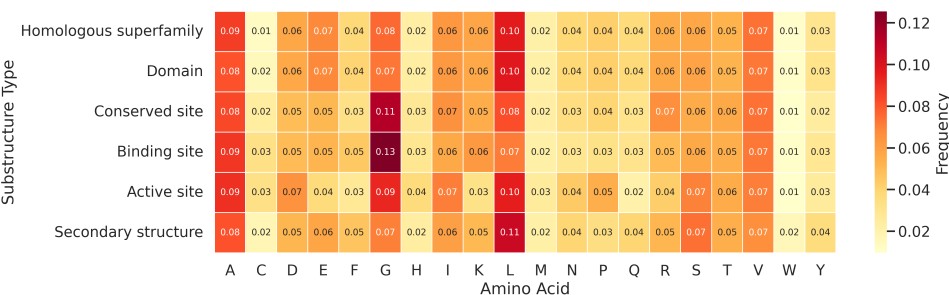

**Figure A.5:** Amino acid composition of substructure types. Each row shows the amino acid composition of that substructure type. Rows sum to 1.

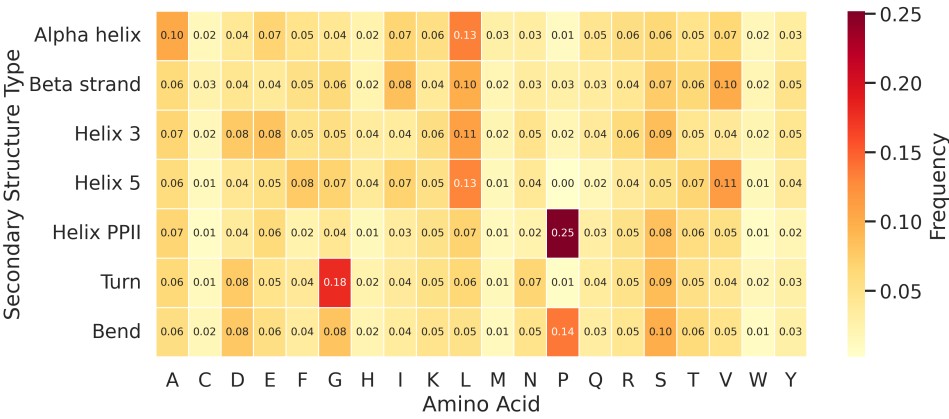

**Figure A.6:** Amino acid composition of granular secondary structure types. Each row shows the amino acid composition of that secondary substructure type. Rows sum to 1.

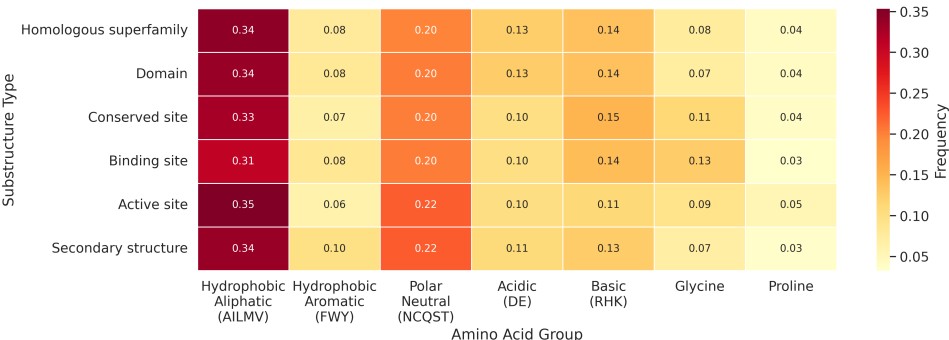

**Figure A.7:** Amino acid composition of substructure types, with amino acids grouped by side chain chemical properties. Each row shows the amino acid composition of that substructure type. Rows sum to 1.

### A.1.3 SUBSTRUCTURE DATASET FILTERING

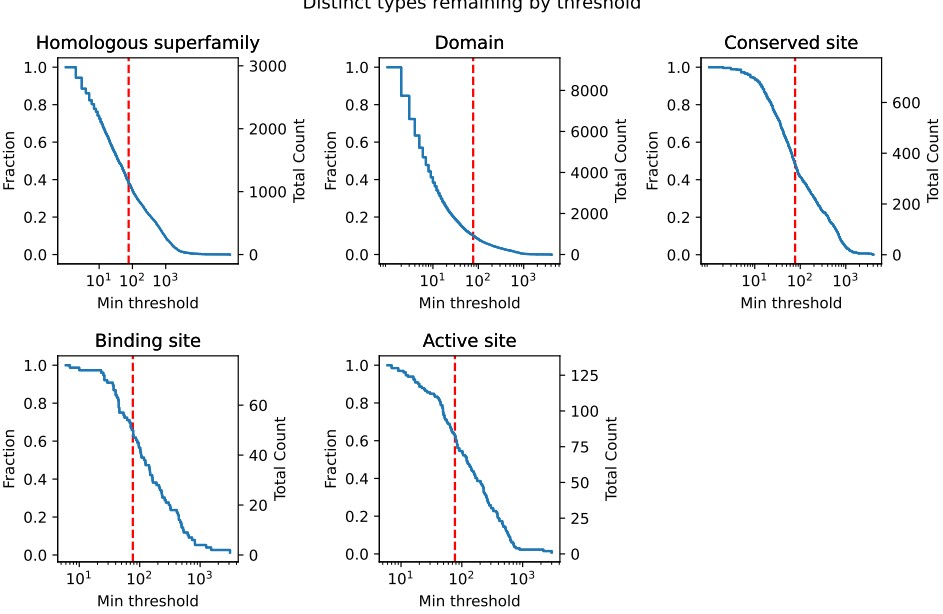

**Figure A.8:** Inverse CDF of **unique types** retained at a given count threshold. The $x$-axis specifies the minimum count for a substructure type to be retained, the left $y$-axis shows the fraction of all unique types retained at the given threshold, and the right $y$-axis shows the absolute count of unique types retained. Facets show different classes of substructural elements. The vertical dashed red lines show the threshold selected for downstream substructure-tuning.

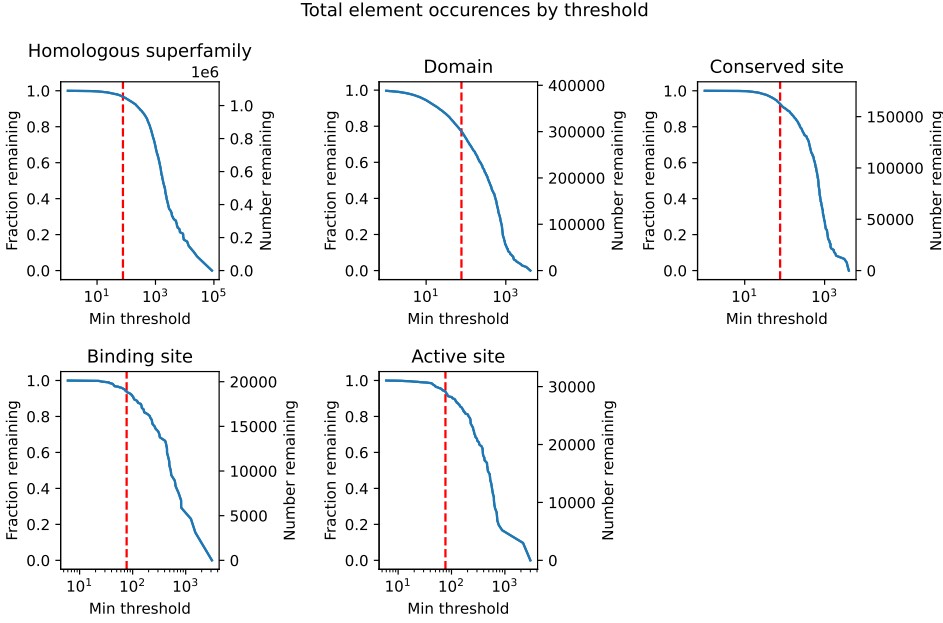

**Figure A.9:** Inverse CDF of **total occurrences** retained at a given count cutoff. This is analogous to Figure A.8 above, but showing total occurrences of substructures rather than unique types, demonstrating that for classes like domains, the majority of annotations come from a small number of domain types.

| Model | EC | GO:BP | GO:CC | GO:MF | Localization (Accuracy) | | Thermostability (Spearman's $\rho$) | Human PPI (AUROC) |
|---|---|---|---|---|---|---|---|---|
| | | | $F_{\max}$ | | Binary | Subcellular | | |
| ESM-C 300M | 0.688 | 0.307 | 0.416 | 0.429 | 0.871 | 0.703 | 0.648 | 0.917 |
| +ST (original, $\geq$ 75) | 0.761 | 0.325 | 0.403 | 0.488 | 0.879 | 0.681 | 0.660 | 0.933 |
| +ST ($\geq$ 25) | 0.802 | 0.32 | 0.399 | 0.526 | 0.851 | 0.693 | 0.64 | 0.917 |
| +ST ($\geq$ 10) | 0.792 | 0.327 | 0.401 | 0.514 | 0.89 | 0.728 | 0.64 | 0.852 |

**Table A.1: Effect of substructure frequency cutoff on protein-level task performance.** The performance of substructure-tuning is robust to the presence of rare substructures. Values in parentheses indicate the minimum number of times a substructure type must occur in the overall SwissProt dataset to be included for substructure-tuning.

| Substructure type | Full SwissProt set | Count $\geq$ 75 | Count $\geq$ 25 | Count $\geq$ 10 |
|---|---|---|---|---|
| Homologous superfamily | 3511 | 1133 | 1685 | 2159 |
| Domain | 15868 | 917 | 1964 | 3506 |
| Conserved site | 748 | 356 | 573 | 691 |
| Binding site | 76 | 48 | 72 | 74 |
| Active site | 133 | 82 | 114 | 127 |

**Table A.2: Number of substructure types included in the dataset at different count cutoffs.** Reducing the cutoff down to 10 greatly expands the number of substructure types included in the dataset.

| Model | EC | GO:BP | GO:CC | GO:MF | Localization (Accuracy) | | Thermostability (Spearman's $\rho$) | Human PPI (AUROC) | FLIP_bind ($F_{max}$) | Biolip binding (AUROC) | Biolip catalytic (AUROC) |
|---|---|---|---|---|---|---|---|---|---|---|---|
| | | $F_{max}$ | | | Binary | Subcellular | | | | | |
| ESM-C 300M | 0.688 | 0.307 | 0.416 | 0.429 | 0.871 | 0.703 | 0.648 | 0.917 | 0.367 | 0.851 | 0.923 |
| +ST (original) | 0.761 | 0.325 | 0.403 | 0.488 | 0.879 | 0.681 | 0.660 | 0.933 | 0.411 | 0.866 | 0.910 |
| +ST (exact match) | 0.789 | 0.317 | 0.403 | 0.5 | 0.88 | 0.687 | 0.647 | 0.892 | 0.372 | 0.847 | 0.879 |
| +ST (similar seq) | 0.777 | 0.317 | 0.385 | 0.507 | 0.83 | 0.686 | 0.645 | 0.889 | 0.375 | 0.849 | 0.912 |

**Table A.3: Effect of stringent data split on substructure-tuning performance.**

### A.1.4 DATASET LEAKAGE ANALYSES

It's generally standard practice when training protein models to consider the dataset used for self-supervised learning as distinct from the downstream evaluation sets, presumably this is because the sequence or structure data is considered sufficiently far removed from the labels used in downstream evaluations (e.g. GO terms, experimental fitness values, etc.). However, substructural annotations are different from sequence or global structure in that they can be more directly tied to protein function. For this reason, we performed a series of analyses to understand whether any of the effects of substructure-tuning could be attributed to data leakage between the substructure-tuning training set and the test sets of any of the evaluation benchmarks.

First, we performed a direct experiment by constructing two versions of more stringent dataset splits:

- "Exact match": To construct this split, we identified all proteins in the Magneton train set that were also contained in the test split of any evaluation benchmark, based on UniProt ID. We then removed all such proteins and all of their corresponding AFDB50 clusters from the Magneton train set. This removed 31,191 of the 423,885 proteins in the Magneton training split.

- "Similar seq": To construct an even more stringent split, we collected the amino acid sequences of all proteins in the test split of any evaluation benchmark, and aligned these sequences to the proteins in the Magneton train set. We then removed from the Magneton train set any protein with greater than 30% sequence similarity and 80% overlap with any evaluation test set protein. Alignments were performed using MMseqs2 (Steinegger & Söding, 2017; Kallenborn et al., 2025) with sequence similarity defined using the `fident` output (command: `mmseqs easy-search eval_seqs.fa train_seqsDB tmp output.tsv --cov-mode 1 -c 0.8 --alignment-mode 3`). By constructing the dataset split in this manner, we ensure that no protein contained in the Magneton train set has more than 30% sequence similarity to any protein in any of the evaluation benchmark test sets (Figure A.10). This process removed 131,440 of the 423,885 proteins in the Magneton training split.

We then performed substructure-tuning of ESM-C 300M using active, binding, and conserved site annotations with each of these stringent training splits and evaluated the resulting models on our evaluation suite (Table A.3). We found that benchmark performance was largely similar to that achieved with the original training, providing evidence that data leakage does not drive any of the effects of substructure-tuning.

As a further step, we leveraged mappings from Gene Ontology terms to InterPro entries[6] to check whether substructure-tuning's improvement on GO MF prediction could be attributed to this mapping. In particular, we checked if the improvement in predicting a GO MF term after substructure-tuning correlated with the representation of its corresponding domains in the Magneton train set. We found that the performance improvement for predicting a given GO MF does not depend on its corresponding domain's representation in the Magneton train set, again supporting the lack of data leakage (Figure A.11, Table A.4).

---

[6] https://www.ebi.ac.uk/GOA/InterPro2GO

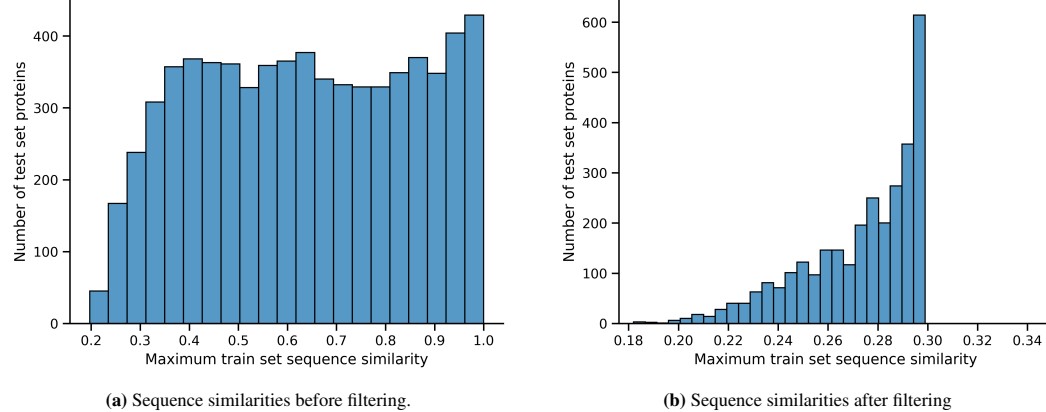

**(a)** Sequence similarities before filtering.

**(b)** Sequence similarities after filtering

**Figure A.10:** Distribution of sequence similarities (`fident` from MMseqs2) between proteins in the test set of any evaluation benchmark and proteins in the Magneton training set. Plotted values are the maximum similarity to any Magneton training set protein.

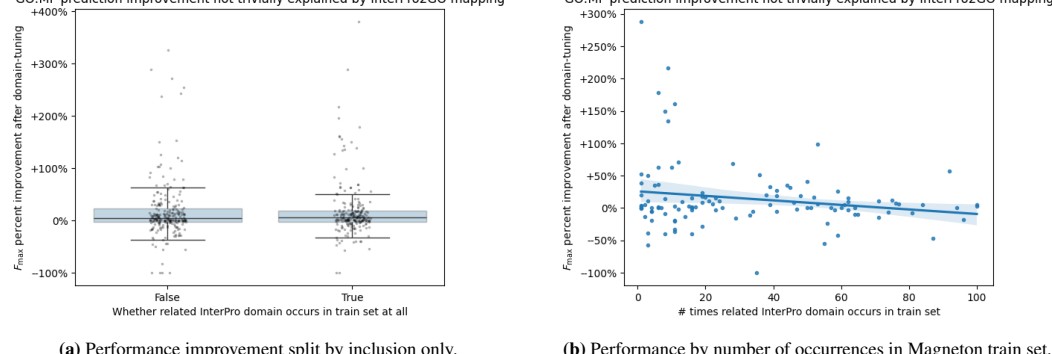

**(a)** Performance improvement split by inclusion only.

**(b)** Performance by number of occurrences in Magneton train set.

**Figure A.11:** GO:MF prediction improvement by representation of associated domain in the Magneton train set.

| | % $F_{\max}$ improvement after substructure-tuning | | | |
|---|---|---|---|---|
| | Mean | 25th percentile | Median | 75th percentile |
| Domains in train set | +17.82% | -2.36% | +5.11% | +18.66% |
| Domains not in train set | +16.84% | -3.40% | +4.88% | +23.35% |

**Table A.4: GO:MF prediction improvement by domain presence in training set.** Improvement in predicting a GO:MF term after substructure-tuning does not correlate with the representation of its corresponding domains in the train set.

### A.1.5 MAGNETON EVALUATION DATASETS

Magneton contains eleven different benchmarking datasets, comprising 14 evaluation tasks. These evaluation tasks represent years of work from the scientific community, both in their original generation and later processing that we build upon, and we acknowledge and thank all of those involved. The included tasks are:

- **Human PPI prediction.** The goal of this task is to predict whether or not two proteins form an interacting pair. This is a binary classification task where the input is two proteins and the output is a binary label indicating interaction or no interaction. The evaluation metric is accuracy. The original dataset is sourced from Pan et al. (2010). We build off of processed data files from Su et al. (2023).

- **Gene Ontology and Enzyme Commission prediction.** The goal of these tasks are to predict the Gene Ontology (GO) or Enzyme Commission (EC) annotations for a protein. There are three categories of GO annotations: Molecular Function (MF), Cellular Component (CC), and Biological Process (BP), each of which captures a different aspect of protein biology and is treated as a separate benchmarking task, giving a total of four tasks. These are multilabel classification tasks where a protein can have multiple annotations. The evaluation metric is $F_{\max}$, the maximum $F_1$ score over possible thresholds. We source original data from Gligorijević et al. (2021).

- **Subcellular and binary localization.** The goal of these tasks is to predict a protein's localization either within multiple cellular compartments (subcellular localization) or whether the protein is membrane-bound or soluble (binary localization). The input is a single protein and this is either a multiclass or binary classification task. The evaluation metric is accuracy. The original dataset is sourced from Almagro Armenteros et al. (2017). We build off of processed data files from Su et al. (2023).

- **Thermostability.** The goal of this task is to predict the stability of a protein under extreme temperatures. The output is a continuous value indicating the thermostability, and the goal is to rank-order proteins according to their experimental values. The evaluation metric is Spearman rank correlation (Spearman's $\rho$) calculated against the experimental values. The original dataset is sourced from Rao et al. (2019).

- **Binding residue categorization.** The goal of this task is to predict whether a given residue binds three different types of ligands: metal ions, small molecules, or nucleic acids. This is a residue-level multilabel classification task. The evaluation metric is $F_{\max}$. The original dataset is sourced from Dallago et al. (2021).

- **Binding and catalytic site prediction.** The goal of these tasks are to predict whether a given residue is part of an annotated binding or catalytic site. This is a residue-level binary classification task. The evaluation metric is AUROC. The original datasets are sourced from experimentally determined structures curated by Zhang et al. (2024a). We build off of processed data files from Yuan et al. (2025).

- **Contact prediction.** The goal of this task is to predict whether two residues within the same protein are "in contact" with each other, which is defined as having alpha-carbon atoms within 8 angstroms of each other in the tertiary structure. The input is a single protein of length $L$ and the output is a $L \times L$ contact map, where element $i, j$ of the contact map is the predicted probability that residues $i$ and $j$ are in contact. The evaluation metric is Precision@L, which calculates precision over the top $L$ most confident contact predictions where $L$ is the protein length. This metric is further stratified into short, medium, and long-range contacts in which the possible residue pairs considered are those whose pairwise separation along the primary sequence is either in $[6, 10]$, $[12, 22]$, or $[24, L]$, respectively. The underlying data are experimentally determined structures from PDB, originally curated by Rao et al. (2019).

- **Zero-shot DMS variant effect prediction**. The goal of this task is to predict the effect of a single or multiple amino acid mutations on a protein's function. The input is the mutated sequence and the output is a continuous value representing fitness. The evaluation metric is Spearman's $\rho$ against the experimentally-determined fitness values from deep mutational scans (DMS) experiments. These tasks are zero-shot in that no supervised training for

variant effect prediction is performed. For each model, we use the author's recommended methods for VEP. The data is sourced from Notin et al. (2023).

For all datasets, protein structures are sourced from AlphaFold DB (Varadi et al., 2022) unless otherwise specified. Accounting of samples per split in each dataset are available in Table A.5.

| Task | Train | Validation | Test | Number of classes | Task type |
|---|---|---|---|---|---|
| EC | 14,466 | 1,599 | 1,715 | 538 | Multilabel |
| GO:BP | 21,470 | 2,393 | 3,394 | 1,943 | Multilabel |
| GO:CC | 9,793 | 1,118 | 3,394 | 320 | Multilabel |
| GO:MF | 22,621 | 2,495 | 3,394 | 489 | Multilabel |
| Subcellular localization | 8,741 | 2,190 | 2,744 | 10 | Multiclass |
| Binary localization | 5,473 | 1,335 | 1,728 | 2 | Binary |
| Thermostability | 5,020 | 636 | 1,329 | N/A | Regression |
| Binding residue categorization | 890 | 102 | 286 | 3 | Multilabel |
| Binding site prediction | 8,231 | 2,389 | 5,182 | 2 | Binary |
| Catalytic site prediction | 2,856 | 603 | 1,165 | 2 | Binary |
| Contact prediction | 20,653 | 209 | 40 | 2 | Binary |
| Variant effect prediction[1] | N/A | N/A | 217 | N/A | Regression |
| Human PPI prediction[2] | 26,313 | 234 | 180 | 2 | Binary |

**Table A.5:** Dataset sizes (in proteins) and number of classes for each benchmarking task.

[1] This is a zero-shot task, hence the lack of training and validation data. Samples correspond to assays, covering 2.3M mutations.

[2] Samples correspond to *pairs* of proteins rather than individual proteins.

## A.2 Training details

### A.2.1 Substructure classification and tuning

For our substructure prediction modules, we use single hidden-layer MLPs where the dimensionality of the hidden layer matches that of the underlying base model. We extract residue-level embeddings from the final hidden layer of the base model, and use mean pooling across a substructure's constituent residues to construct a single embedding per substructure. When training with multiple categories of substructures, we use a separate prediction module for each category. The training loss is the sum of the classification losses across all categories.

We train using AdamW (Loshchilov & Hutter, 2019) ($\beta_1 = 0.9$, $\beta_2 = 0.999$), learning rates of $10^{-3}$ and dropout rate of 0.1 for the prediction heads, learning rate of $10^{-5}$ for the base model, and EWC weight of 400 (as used by original authors). Training proceeded until convergence of validation loss. All runs used batches of 32 proteins with a variable number of substructures per protein. All training was performed using `bfloat16` on one to four NVIDIA A100 GPUs.

**Elastic weight consolidation.** Briefly, EWC uses the diagonal of the Fisher information matrix $\mathcal{F}$ as weights on a loss that regularizes towards the original model parameters, $\theta_0$:

$$L = L_c(\theta) + \sum_i \frac{\lambda}{2} F_i \left( \theta_i - \theta_{0_i} \right)^2$$

where $L_c$ is the substructure classification loss. $\mathcal{F}$ can be estimated at the beginning of training as the squared gradients of the original loss with respect to the model parameters using the training set. In our case, the original loss corresponds to the training objective of the underlying model (*e.g.* masked amino acid prediction for ESM models, masked amino acid prediction in presence of structure tokens for ProSST or SaProt). In practice, $\mathcal{F}$ is estimated by making a single pass over the training set, running backward passes using the original loss, and averaging the squared gradients over minibatches.

While similar to a $L_2$ loss, EWC has two advantages over a simple $L_2$ or weight decay regularization: (1) weights are decayed towards the original weights of the base model, (2) per-parameter weights are applied which correspond to the importance of that parameter for the original task. We

|  | Frozen base model | Full finetuning | EWC finetuning |
|---|---|---|---|
| Validation MLM loss | 1.24 | 2.30 | 1.75 |

**Table A.6: Masked language modeling loss under different finetuning strategies.**

| Model | Substructures used for tuning | EC | GO:BP | GO:CC | GO:MF | Zero-shot DMS (Spearman's $\rho$) | Binary Localization (Accuracy) | Subcellular Localization (Accuracy) |
|---|---|---|---|---|---|---|---|---|
|  |  | $F_{\max}$ | | | |  |  |  |
| ESM-C (300M) | N/A | 0.688 | 0.307 | 0.416 | 0.429 | 0.432 | 0.871 | 0.703 |
| EWC | Domain | 0.776 | 0.307 | 0.403 | 0.501 | 0.340 | 0.811 | 0.640 |
| No EWC | Domain | 0.831 | 0.311 | 0.386 | 0.531 | TBD | 0.802 | 0.643 |

**Table A.7: Effect of EWC on substructure-tuning performance.** Training with EWC offers a good balance of maintaining improvement on function-related tasks while mitigating performance decreases on other tasks.

selected EWC due to its simplicity and ease of use, as the estimate of $\mathcal{F}$ can be calculated a single time and used for the remainder of training or for other training runs using the same base model, as opposed to alternate methods like replay buffers. For more details, please refer to Kirkpatrick et al. (2017).

Empirically, use of EWC is motivated by an increase in the masked language modeling loss and validation perplexity when performing substructure-tuning as shown in Table A.6 and Figure A.12. We additionally observe that EWC offers a good balance of maintaining improvement on function-related tasks while mitigating performance decreases on other tasks (Table A.7).

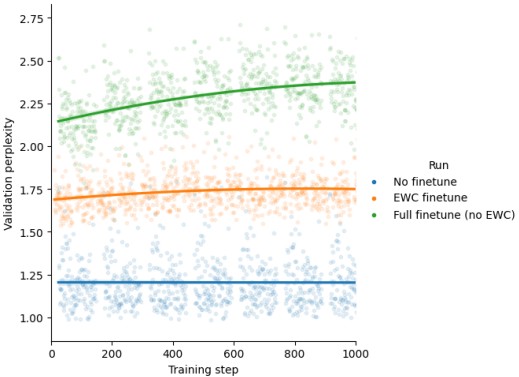

**Figure A.12: Validation perplexity for finetuning ESM-C 300M on domain annotations with and without EWC.**

### A.2.2 DOWNSTREAM TASK TRAINING DETAILS

We use different head models for different scales of supervised downstream tasks:

- **Protein-level.** For protein-level tasks such as GO term prediction, we construct a protein-level representation for each protein following author's recommendations. For ESM-2, we use the final embedding of the `CLS` token as the protein-level embedding. For all other models, we mean pool the final hidden layer representations of all residue tokens (*i.e.* excluding `CLS`, `EOS`, and `PAD` tokens). Prediction heads are then single hidden-layer MLPs with hidden dimensionality matching the hidden dimension of the underlying model.

- **Residue-level.** For residue-level tasks such as binding site prediction, we use a head model consisting of a single 1-dimensional convolutional layer with zero-padding and filter width 5, followed by a nonlinearity and linear layer to the final output.

- **Protein-protein interaction.** For protein-protein interaction prediction, we extract protein-level embeddings as above, concatenate the embeddings for the two input proteins, and pass into a single hidden-layer MLP.

- **Contact prediction**. For contact prediction, we use the `EsmContactPredictionHead` from the `transformers` Python package, which trains a linear regression on top of attention weights from all attention heads in the underlying model.

To train the models, we used AdamW ($\beta_1 = 0.9, \beta_2 = 0.999$) with a learning rate of $10^{-2}$, weight decay of $10^{-2}$, dropout rate of 0.1, and a batch size of 32. Training proceeded for a maximum of 20 epochs, selecting the best model based on validation set performance. When performing full task-specific finetuning, we use a learning rate of $2 \times 10^{-5}$ for the base model and scale the number of GPUs and gradient accumulation steps accordingly to maintain a batch size of 32. All training was performed using `bfloat16` on one to four NVIDIA A100 GPUs.

## A.2.3 TASK-SPECIFIC FINETUNING

| Model | EC | GO:BP | GO:CC | GO:MF | Localization (Accuracy) | | Thermostability (Spearman's $\rho$) |
|---|---|---|---|---|---|---|---|
| | | | $F_{\max}$ | | Binary | Subcellular | |
| ESM2-150M | 0.911 | 0.352 | 0.451 | 0.658 | 0.928 | 0.810 | 0.694 |
| +ST | 0.910 | 0.349 | 0.444 | 0.658 | 0.936 | 0.791 | 0.702 |
| ESM2-650M | 0.910 | 0.356 | 0.446 | 0.662 | 0.933 | 0.824 | 0.703 |
| +ST | 0.914 | 0.360 | 0.457 | 0.665 | 0.930 | 0.793 | 0.703 |
| ESM-C 300M | 0.916 | 0.368 | 0.470 | 0.667 | 0.932 | 0.806 | 0.693 |
| +ST | 0.920 | 0.355 | 0.454 | 0.669 | 0.941 | 0.804 | 0.693 |
| ESM-C 600M | 0.920 | 0.374 | 0.469 | 0.669 | 0.917 | 0.812 | 0.705 |
| +ST | 0.924 | 0.372 | 0.468 | 0.667 | 0.931 | 0.828 | 0.699 |
| ProSST-2048 | 0.911 | 0.319 | 0.439 | 0.631 | 0.912 | 0.760 | 0.673 |
| +ST | 0.901 | 0.336 | 0.427 | 0.638 | 0.923 | 0.744 | 0.670 |

**Table A.8:** Evaluation task performance for models with and without substructure-tuning, and **full finetuning for the downstream task.** In this regime, we find that task-specific finetuning largely results in similar models, showing that imbuing substructural information via supervised finetuning may be brittle in the face of aggressive task-specific finetuning.

## A.2.4 DOWNSTREAM TASK PERFORMANCE UNCERTAINTY ESTIMATES

To assess whether substructure-tuning's effect on downstream task performance, we performed 50 rounds of bootstrapping when calculating the final performance metrics. For both protein-level tasks (A.9) and residue-level tasks (A.10), we found that the performance differences observed were beyond the scale of the uncertainty estimates.

## A.3 MECHANISTIC ANALYSES

### A.3.1 EMBEDDING ANALYSES

We assessed the consistency of substructure representations before and after substructure-tuning by calculating silhouette scores of the resulting embeddings, which capture how tightly grouped the representations are for different occurrences of the same substructure type (Figure A.13). This approach also allows us to compute silhouette scores for substructure types that were completely excluded from substructure-tuning. We additionally explored kNN-based classification as a method to assess zero-shot generalization to unseen substructure types (Figure A.14).

To explore cross-scale consistency of different substructure classes and directly assess the utility of a shared bottleneck, we performed substructure-tuning of ESM-C 300M using a single classification head operating over all of the active, binding, and conserved site substructure types. We found that using a single shared head is consistent with using separate heads in terms of performance boosts in function-centric tasks over the base model, but with varying effects on other tasks (Table A.11).

### A.3.2 GRADIENT CONFLICT ANALYSIS

The gradient conflict analysis was performed as follows:

- Begin with a base ESM-C 300M model (i.e. no substructure-tuning)
- For each evaluation dataset listed in Table A.12, compute the gradient for the associated loss function for 1000 batches of batch size 32.
- For substructure-tuning with active, binding, and conserved-site annotations, compute the gradient for the substructure classification objective for 1000 batches of batch size 32 (where batch size refers to the number of proteins per batch, each of which must contain at least one substructure).
- Compute pairwise similarities between all gradient vectors.

| Model | EC | GO:BP | GO:CC | GO:MF | Localization (Accuracy) | | Thermostability (Spearman's $\rho$) | Human PPI (AUROC) |
|---|---|---|---|---|---|---|---|---|
| | $F_{\max}$ | | | | Binary | Subcellular | | |
| ESM2-150M | $0.727 \pm 0.008$ | $0.316 \pm 0.003$ | $0.416 \pm 0.004$ | $0.441 \pm 0.008$ | $0.869 \pm 0.006$ | $0.694 \pm 0.009$ | $0.627 \pm 0.017$ | $0.933 \pm 0.015$ |
| +ST | $0.742 \pm 0.008$ | $0.323 \pm 0.004$ | $0.415 \pm 0.004$ | $0.473 \pm 0.007$ | $0.866 \pm 0.007$ | $0.679 \pm 0.008$ | $0.582 \pm 0.016$ | $0.919 \pm 0.016$ |
| ESM2-650M | $0.755 \pm 0.005$ | $0.319 \pm 0.003$ | $0.431 \pm 0.004$ | $0.486 \pm 0.007$ | $0.876 \pm 0.009$ | $0.710 \pm 0.009$ | $0.643 \pm 0.017$ | $0.939 \pm 0.018$ |
| +ST | $0.745 \pm 0.007$ | $0.321 \pm 0.003$ | $0.440 \pm 0.004$ | $0.534 \pm 0.007$ | $0.895 \pm 0.009$ | $0.749 \pm 0.008$ | $0.655 \pm 0.018$ | $0.935 \pm 0.018$ |
| ESM-C 300M | $0.688 \pm 0.007$ | $0.307 \pm 0.003$ | $0.416 \pm 0.004$ | $0.429 \pm 0.006$ | $0.871 \pm 0.008$ | $0.703 \pm 0.008$ | $0.648 \pm 0.017$ | $0.917 \pm 0.020$ |
| +ST | $0.761 \pm 0.007$ | $0.325 \pm 0.003$ | $0.403 \pm 0.004$ | $0.488 \pm 0.008$ | $0.879 \pm 0.008$ | $0.681 \pm 0.008$ | $0.656 \pm 0.018$ | $0.933 \pm 0.015$ |
| ESM-C 600M | $0.701 \pm 0.005$ | $0.312 \pm 0.003$ | $0.403 \pm 0.003$ | $0.436 \pm 0.006$ | $0.863 \pm 0.008$ | $0.713 \pm 0.007$ | $0.668 \pm 0.016$ | $0.927 \pm 0.020$ |
| +ST | $0.780 \pm 0.008$ | $0.319 \pm 0.003$ | $0.385 \pm 0.005$ | $0.527 \pm 0.007$ | $0.872 \pm 0.008$ | $0.635 \pm 0.009$ | $0.667 \pm 0.013$ | $0.902 \pm 0.022$ |
| SaProt (650M) | $0.778 \pm 0.007$ | $0.326 \pm 0.004$ | $0.453 \pm 0.005$ | $0.538 \pm 0.006$ | $0.887 \pm 0.009$ | $0.784 \pm 0.007$ | $0.692 \pm 0.016$ | $0.952 \pm 0.015$ |
| +ST | $0.839 \pm 0.005$ | $0.339 \pm 0.004$ | $0.446 \pm 0.004$ | $0.584 \pm 0.007$ | $0.896 \pm 0.006$ | $0.741 \pm 0.008$ | $0.697 \pm 0.015$ | $0.932 \pm 0.017$ |
| ProSST-2048 | $0.778 \pm 0.007$ | $0.317 \pm 0.003$ | $0.426 \pm 0.004$ | $0.522 \pm 0.006$ | $0.878 \pm 0.005$ | $0.693 \pm 0.008$ | $0.686 \pm 0.014$ | $0.925 \pm 0.024$ |
| +ST | $0.791 \pm 0.006$ | $0.314 \pm 0.004$ | $0.420 \pm 0.004$ | $0.567 \pm 0.007$ | $0.853 \pm 0.010$ | $0.683 \pm 0.009$ | $0.648 \pm 0.017$ | $0.883 \pm 0.023$ |

**Table A.9: Protein-level task performance for base models and models with substructure-tuning (+ST).** Values shown as mean $\pm$ standard deviation. Mean and std dev computed from 50 runs of bootstrapping.

| Model | Binding residue ($F_{\max}$) | Functional site prediction | |
|---|---|---|---|
| | | Binding (AUROC) | Catalytic |
| ESM2-150M | $0.379 \pm 0.005$ | $0.871 \pm 0.001$ | $0.910 \pm 0.002$ |
| +ST | $0.327 \pm 0.005$ | $0.852 \pm 0.001$ | $0.890 \pm 0.002$ |
| ESM2-650M | $0.366 \pm 0.006$ | $0.849 \pm 0.001$ | $0.912 \pm 0.002$ |
| +ST | $0.362 \pm 0.006$ | $0.851 \pm 0.001$ | $0.927 \pm 0.002$ |
| ESM-C 300M | $0.367 \pm 0.005$ | $0.851 \pm 0.001$ | $0.923 \pm 0.002$ |
| +ST | $0.411 \pm 0.006$ | $0.866 \pm 0.001$ | $0.910 \pm 0.003$ |
| ESM-C 600M | $0.357 \pm 0.006$ | $0.850 \pm 0.001$ | $0.921 \pm 0.002$ |
| +ST | $0.368 \pm 0.006$ | $0.852 \pm 0.001$ | $0.906 \pm 0.002$ |
| SaProt (650M) | $0.423 \pm 0.007$ | $0.891 \pm 0.001$ | $0.923 \pm 0.007$ |
| +ST | $0.400 \pm 0.005$ | $0.871 \pm 0.001$ | $0.924 \pm 0.002$ |
| ProSST-2048 | $0.375 \pm 0.006$ | N/A | N/A |
| +ST | $0.342 \pm 0.006$ | N/A | N/A |

**Table A.10: Residue-level task performance for base models and models with substructure-tuning (+ST).** Values shown as mean $\pm$ standard deviation. Mean and std dev computed from 50 runs of bootstrapping.

The final results of this analysis are shown for comparison of gradient vectors derived from batches from the same task (Table A.12a) and from batches from different tasks (Table A.12b).

### A.3.3 SUBSTRUCTURE ATTRIBUTION ANALYSIS

To understand correlation of substructure-tuned embeddings with known functional motifs, we performed an explainability analysis by first identifying all proteins in the GO:MF test set whose GO:MF annotations could be attributed to a specific domain (accomplished via InterPro2GO mappings), yielding 81 (protein, domain, GO:MF) triplets. For each protein, we computed a saliency map to calculate residue contributions to the prediction of the associated GO:MF term. We then calculated the total contribution of residues contained within the associated domain and normalized this by the total contribution across all residues in the protein. For ESM-C 300M before and after substructure-tuning, we find that attributions to domain residues increased by an average of 17%, indicating increased usage of substructures for function prediction after substructure-tuning (Appendix Figure A.15).

As a concrete example, we provide a saliency map for the protein Q15436 and the prediction of its annotated GO:0008270 term. The GO:0008270 GO term can be directly mapped to the InterPro domain IPR006895 contained at positions 58 to 98 in the protein. We can observe that the untuned

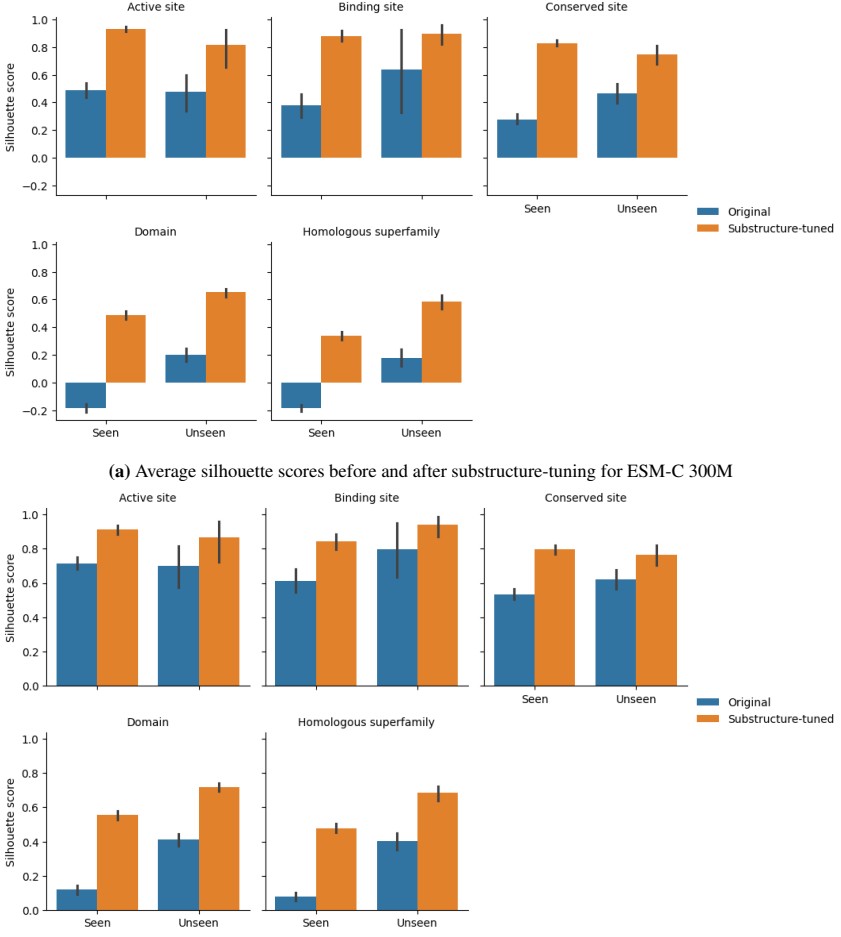

**(a)** Average silhouette scores before and after substructure-tuning for ESM-C 300M

**(b)** Average silhouette scores before and after substructure-tuning for SaProt 650M

**Figure A.13: Average silhouette scores of each substructure type class, before and after substructure-tuning, for both seen and unseen classes.** Despite never training on any examples of the unseen substructure types, substructure-tuning results in more consistent representations of these substructures.

| Model | EC | GO:BP | GO:CC | GO:MF | Localization (Accuracy) | | Thermostability (Spearman's $\rho$) | Human PPI (AUROC) | FLIP_bind ($F_{max}$) | Biolip binding (AUROC) | Biolip catalytic (AUROC) |
| | | $F_{max}$ | | | Binary | Subcellular | | | | | |
|---|---|---|---|---|---|---|---|---|---|---|---|
| ESM-C 300M | 0.688 | 0.307 | 0.416 | 0.429 | 0.871 | 0.703 | 0.648 | 0.917 | 0.367 | 0.851 | 0.923 |
| +ST (original, separate heads) | 0.761 | 0.325 | 0.403 | 0.488 | 0.879 | 0.681 | 0.660 | 0.933 | 0.411 | 0.866 | 0.910 |
| +ST (single head) | 0.777 | 0.304 | 0.392 | 0.513 | 0.872 | 0.704 | 0.666 | 0.887 | 0.39 | 0.85 | 0.888 |

**Table A.11: Effect of using a single shared classification head for substructure-tuning.** The effect of using a single shared head is consistent with using separate heads in terms of performance boosts in function-centric tasks over the base model, but with varying effects on other tasks.

model exhibits less attribution to the relevant domain (Appendix Figure A.16) than the substructure-tuned model (Appendix Figure A.17).

## A.4 ABLATIONS

### A.4.1 ALTERNATE POOLING TECHNIQUES

We initially developed our approach using mean pooling to construct substructure representations due to its simplicity, but also explored additional aggregation techniques to help expand the completeness of our work. We note that the choice of a simple pooling over a learnable aggregation was intentional: since the goal of substructure-tuning is to use the supervised substructure classification

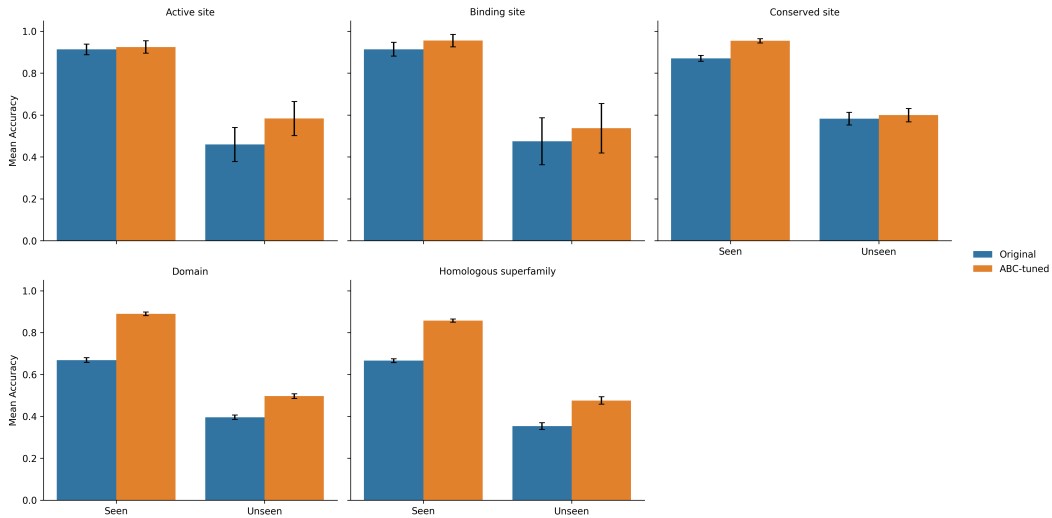

**Figure A.14: Average accuracy of kNN-based classification of each substructure class using embeddings, before and after substructure-tuning, for both seen and unseen classes.**

| Task | GO:MF | EC | Binding residue | Functional site prediction (binding) | Functional site prediction (catalytic) | Substructure classification |
|---|---|---|---|---|---|---|
| Gradient cosine similarity | $0.957 \pm 0.004$ | $0.956 \pm 0.004$ | $0.961 \pm 0.007$ | $0.964 \pm 0.004$ | $0.973 \pm 0.022$ | $0.095 \pm 0.077$ |

(a) Within task gradient similarity (mean $\pm$ std deviation)

| Task | GO:MF | EC | Binding residue | Functional site prediction (binding) | Functional site prediction (catalytic) |
|---|---|---|---|---|---|
| Gradient cosine similarity | $-0.006 \pm 0.047$ | $0.003 \pm 0.027$ | $0.022 \pm 0.056$ | $0.002 \pm 0.055$ | $-0.005 \pm 0.038$ |

(b) Task gradient similarity with substructure classification gradient (mean $\pm$ std deviation)

**Table A.12: Gradient conflict analysis for substructure-tuning.** These results suggest that the task-specific effects of substructure-tuning is not due to a simple global misalignment between the substructure objective and certain downstream tasks.

problem to tune the parameters of the underlying protein model, we wanted to encourage learning to occur within the base model rather than in a parameterized pooling module.

To investigate this, we performed substructure-tuning using max pooling or a learnable attention pooling. Both of these experiments performed substructure-tuning on ESM-C 300M using active, binding, and conserved site annotations. We find that substructure-tuning is robust to the exact method used for pooling residue-level embeddings to substructural embeddings (Appendix Table A.13).

### A.4.2 ADDITIONAL MODELS

We additionally conducted a direct comparison of substructure-tuning to methods that use global structure to tune sequence models would help contextualize our work. To ensure a fair comparison, we have integrated ESM-S into Magneton and evaluated it on our benchmark suite (Appendix Table A.14).

We find that substructure-tuning compares favorably to ESM-S, with generally larger improvements on function-related tasks and smaller performance drop-offs in other tasks. We also note that substructure-tuning is also useful for models that already incorporate global structure information, something which has not been shown for ESM-S's methodology.

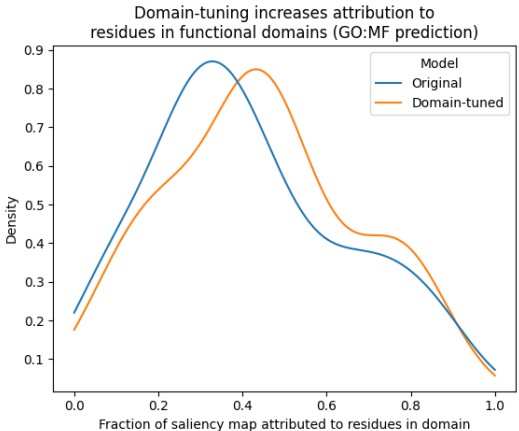

**Figure A.15: Saliency map indicating attributions to domain residues during GO:MF predictions before and after domain-tuning.**

M T T Y L E F I Q Q N E E R D G V R F S W N V W P S S R L E A T R M V V P V A
A L F T P L K E R P D L P P I Q Y E P [DOM_START] V L C S R T T C R A V L N P L
C Q V D Y R A K L W A C N F C Y Q R N Q F P P S Y [DOM_END] A G I S E L N Q P
A E L L P Q F S S I E Y V V L R G P Q M P L I F L Y V V D T C M E D E D L Q A L
K E S M Q M S L S L L P P T A L V G L I T F G R M V Q V H E L G C E G I S K S Y
V F R G T K D L S A K Q L Q E M L G L S K V P L T Q A T R G P Q V Q Q P P P S
N R F L Q P V Q K I D M N L T D L L G E L Q R D P W P V P Q G K R P L R S S G
V A L S I A V G L L E C T F P N T G A R I M M F I G G P A T Q G P G M V V G D
E L K T P I R S W H D I D K D N A K Y V K K G T K H F E A L A N R A A T T G H
V I D I Y A C A L D Q T G L L E M K C C P N L T G G Y M V M G D S F N T S L F
K Q T F Q R V F T K D M H G Q F K M G F G G T L E I K T S R E I K I S G A I G P
C V S L N S K G P C V S E N E I G T G G T C Q W K I C G L S P T T T L A I Y F E
V V N Q H N A P I P Q G G R G A I Q F V T Q Y Q H S S G Q R R I R V T T I A R N
W A D A Q T Q I Q N I A A S F D Q E A A A I L M A R L A I Y R A E T E E G P D
V L R W L D R Q L I R L C Q K F G E Y H K D D P S S F R F S E T F S L Y P Q F M
F H L R R S S F L Q V F N N S P D E S S Y Y R H H F M R Q D L T Q S L I M I Q P
I L Y A Y S F S G P P E P V L L D S S S I L A D R I L L M D T F F Q I L I Y H G E
T I A Q W R K S G Y Q D M P E Y E N F R H L L Q A P V D D A Q E I L H S R F P
M P R Y I D T E H G G S Q A R F L L S K V N P S Q T H N N M Y A W G Q E S G A
P I L T D D V S L Q V F M D H L K K L A V S S A A

**Figure A.16: Example saliency map before substructure-tuning.** [DOM_START] and [DOM_END] denote the beginning and end of the domain within the sequence respectively.

M T T Y L E F I Q Q N E E R D G V R F S W N V W P S S R L E A T R M V V P V A
A L F T P L K E R P D L P P I Q Y E P [DOM_START] V L C S R T T C R A V L N P L
C Q V D Y R A K L W A C N F C Y Q R N Q F P P S Y [DOM_END] A G I S E L N Q P
A E L L P Q F S S I E Y V V L R G P Q M P L I F L Y V V D T C M E D E D L Q A L
K E S M Q M S L S L L P P T A L V G L I T F G R M V Q V H E L G C E G I S K S Y
V F R G T K D L S A K Q L Q E M L G L S K V P L T Q A T R G P Q V Q Q P P P S
N R F L Q P V Q K I D M N L T D L L G E L Q R D P W P V P Q G K R P L R S S G
V A L S I A V G L L E C T F P N T G A R I M M F I G G P A T Q G P G M V V G D
E L K T P I R S W H D I D K D N A K Y V K K G T K H F E A L A N R A A T T G H
V I D I Y A C A L D Q T G L L E M K C C P N L T G G Y M V M G D S F N T S L F
K Q T F Q R V F T K D M H G Q F K M G F G G T L E I K T S R E I K I S G A I G P
C V S L N S K G P C V S E N E I G T G G T C Q W K I C G L S P T T T L A I Y F E
V V N Q H N A P I P Q G G R G A I Q F V T Q Y Q H S S G Q R R I R V T T I A R N
W A D A Q T Q I Q N I A A S F D Q E A A A I L M A R L A I Y R A E T E E G P D
V L R W L D R Q L I R L C Q K F G E Y H K D D P S S F R F S E T F S L Y P Q F M
F H L R R S S F L Q V F N N S P D E S S Y Y R H H F M R Q D L T Q S L I M I Q P
I L Y A Y S F S G P P E P V L L D S S S I L A D R I L L M D T F F Q I L I Y H G E
T I A Q W R K S G Y Q D M P E Y E N F R H L L Q A P V D D A Q E I L H S R F P
M P R Y I D T E H G G S Q A R F L L S K V N P S Q T H N N M Y A W G Q E S G A
P I L T D D V S L Q V F M D H L K K L A V S S A A

**Figure A.17: Example saliency map after substructure-tuning.** [DOM_START] and [DOM_END] denote the beginning and end of the domain within the sequence respectively.

| Model | EC | GO:BP | GO:CC | GO:MF | Localization (Accuracy) | | Thermostability (Spearman's $\rho$) | Human PPI (AUROC) |
|---|---|---|---|---|---|---|---|---|
| | | | $F_{max}$ | | Binary | Subcellular | | |
| ESM-C 300M | 0.688 | 0.307 | 0.416 | 0.429 | 0.871 | 0.703 | 0.648 | 0.917 |
| +ST (original, mean) | 0.761 | 0.325 | 0.403 | 0.488 | 0.879 | 0.681 | 0.660 | 0.933 |
| +ST (max) | 0.777 | 0.304 | 0.392 | 0.513 | 0.872 | 0.704 | 0.666 | 0.887 |
| +ST (attention) | 0.778 | 0.317 | 0.392 | 0.505 | 0.815 | 0.675 | 0.643 | 0.89 |

**Table A.13: Effect of pooling method on substructure-tuning performance.** Substructure-tuning is robust to the exact method used for pooling residue-level embeddings to substructural embeddings.

| Model | EC | GO:BP | GO:CC | GO:MF | Localization (Accuracy) | | Thermostability (Spearman's $\rho$) | Human PPI (AUROC) | Binding residue ($F_{max}$) | Functional site prediction | |
|---|---|---|---|---|---|---|---|---|---|---|---|
| | | | $F_{max}$ | | Binary | Subcellular | | | | Binding | Catalytic (AUROC) |
| ESM2-150M | 0.727 | 0.316 | 0.416 | 0.441 | 0.869 | 0.694 | 0.627 | 0.933 | 0.379 | 0.871 | 0.910 |
| +ST | 0.742 | 0.324 | 0.415 | 0.473 | 0.866 | 0.679 | 0.582 | 0.919 | 0.327 | 0.852 | 0.890 |
| ESM-S 150M | 0.760 | 0.315 | 0.373 | 0.460 | 0.866 | 0.677 | 0.631 | 0.917 | 0.352 | 0.871 | 0.885 |
| ESM2-650M | 0.755 | 0.319 | 0.431 | 0.486 | 0.876 | 0.710 | 0.643 | 0.939 | 0.366 | 0.849 | 0.912 |
| +ST | 0.745 | 0.321 | 0.440 | 0.534 | 0.895 | 0.749 | 0.655 | 0.935 | 0.362 | 0.851 | 0.927 |
| ESM-S 650M | 0.794 | 0.327 | 0.422 | 0.497 | 0.896 | 0.679 | 0.653 | 0.912 | 0.363 | 0.876 | 0.902 |
| S-PLM (650M) | 0.566 | 0.284 | 0.375 | 0.482 | 0.910 | 0.622 | 0.682 | 0.911 | 0.358 | 0.500 | 0.500 |

**Table A.14: Comparison of substructure-tuning with global structure tuning methods.** Substructure-tuning compares favorably to ESM-S and S-PLM, with generally larger improvements on function-related tasks and similar performance drop-offs in other tasks.

