# OpenReview forum: "Greater than the Sum of Its Parts:  Building Substructure into Protein Encoding Models"
_ICLR.cc/2026/Conference — ICLR 2026 Poster_

### Official Review · Reviewer_w8Xw · 2025-10-27

**Soundness:** 3
**Presentation:** 3
**Contribution:** 2
**Rating:** 6
**Confidence:** 4

**Summary:**

The paper introduces Magneton, an integrated environment for developing substructure-aware protein representation models. Magneton comprises three components:

(1) a large-scale dataset of 530 k proteins annotated with 1.7 M substructures spanning 13 k types;

(2) a training framework that incorporates these substructures into existing sequence- or structure-based encoders; and

(3) a benchmark suite of 13 tasks covering residue-, substructure-, protein-, and interaction-level evaluations.

Building upon this environment, the authors propose substructure-tuning, a supervised fine-tuning strategy that distills substructural information into pretrained protein models. Experiments across six state-of-the-art encoders (150 M–650 M parameters) show consistent ≈5 % improvements in function-related tasks (e.g., EC and GO prediction) and demonstrate that substructural signals complement global structural information.

**Strengths:**

1. The paper identifies an under-explored gap—modern protein language models neglect conserved substructures, despite their well-known biological importance. The proposed Magneton environment and “substructure-tuning” paradigm offer a novel, systematic way to integrate decades of curated biological knowledge into modern representation learning.

2. The dataset construction and methodology are rigorous. The authors curate and filter substructures from major biological databases (Pfam, InterPro, DSSP, SwissProt), perform consistent train/validation/test splitting, and benchmark on 13 tasks. The experimental setup spans multiple model architectures and modalities, lending credibility to the reported gains.

3. The manuscript is well-written and logically structured. Figures 1–3 clearly illustrate the motivation, architecture, and effects of substructure-tuning.

4. The work lays a foundation for multi-scale protein modeling by linking residue-, motif-, and domain-level representations. This contribution is both scientifically meaningful and practically useful, likely to influence future developments in protein representation learning, function prediction, and design.

**Weaknesses:**

1. Substructure information is only injected through supervised fine-tuning; the model architecture itself remains unchanged. As the authors note, this approach can be brittle under task-specific fine-tuning and may not fully exploit hierarchical dependencies. Exploring architectural biases (e.g., graph-hierarchical encoders) would strengthen the contribution.

2. The dataset filters out substructures with < 75 occurrences. While this simplifies training, it biases learning toward frequent structural motifs and limits generalization to rare but functionally critical substructures.

3. Substructure-tuning improves global functional prediction but sometimes degrades residue-level metrics. This trade-off warrants deeper analysis—e.g., are low-level features being overwritten, or are substructure objectives misaligned with certain downstream signals?

4. The paper could better contextualize gains against concurrent structure-aware fine-tuning strategies (e.g., S-PLM, ESM-S). A direct quantitative comparison or ablation with those would clarify the incremental benefit of substructure-tuning.

5. While performance tables are comprehensive, confidence intervals or variance over multiple runs are missing, which is important given modest absolute improvements.

**Questions:**

1. Could the authors discuss whether hierarchical or graph-based encoders (e.g., multi-resolution GNNs) might integrate substructures more stably than fine-tuning alone? A comparison would help isolate the source of gains.

2. Have the authors evaluated whether the tuned models can generalize to rare or unseen motifs not present in the training subset? Such analysis would demonstrate the learned “substructure awareness” beyond supervised classes.

3. Since different substructure classes use separate heads, how correlated are their learned embeddings? Could joint attention or shared bottlenecks yield better cross-scale consistency?

4. One potential advantage of substructure-aware representations is improved interpretability. Can the authors provide examples where substructure-tuned embeddings correlate with known functional motifs or mechanistic insight?

---

> ### Author Response · Authors · 2025-11-21
> **Response to reviewer w8Xw (2/2)**
>
> > W1 & Q1: Alternate architectures such as hierarchical or graph-based encoders
>
> This is a great point and we agree that the fact that substructure-tuning is brittle in the face of aggressive task-specific finetuning points towards architectural modifications such as hierarchical encoders as the next step for taking advantage of substructural information. We are very excited about developing novel model architectures that explicitly incorporate substructure-aware inductive biases and are currently exploring this direction. This work focused on developing the dataset and environment for developing substructure-aware models and exploring lightweight methods to infuse substructure information directly into existing protein models. We discuss alternate architectures briefly in our “Conclusion and Future Work” section, but have expanded the discussion to make the connection to hierarchical architectures more explicit.
>
> > Q2: whether the tuned models can generalize to rare or unseen motifs not present in the training subset?
>
> We appreciate this question, as it prompted us to more deeply investigate the effects of substructure-tuning on unseen substructures. We found that **substructure-tuning encourages models to learn general features of functional substructures, beyond just signatures of specific substructure types** and that these features **generalize to rare and unseen substructures**. For a description of results and analysis, please see point P2 in the global response.
>
> > Q3: Since different substructure classes use separate heads, how correlated are their learned embeddings? Could joint attention or shared bottlenecks yield better cross-scale consistency?
>
> This is an insightful question and suggests interesting directions regarding how to best model functional substructures within proteins at different scales. To directly assess the utility of a shared bottleneck, we trained a model that used a single classification head operating over all of the active, binding, and conserved site substructure types:
>
> |Model|EC ($F_\max$)|GO:BP ($F_\max$)|GO:CC ($F_\max$)|GO:MF ($F_\max$)|Binary Localization (Accuracy)|Subcellular Localization (Accuracy)|Thermostability (Spearman's rho)|Human PPI (AUROC)|FLIP_bind ($F_\max$)|Biolip binding (AUROC)|Biolip catalytic (AUROC)|
> |---|---|---|---|---|---|---|---|---|---|---|---|
> |ESM-C 300M|0.688|0.307|0.416|0.429|0.871|0.703|0.648|0.917|0.367|0.851|0.923|
> |+ST (original, separate heads)|0.761|0.325|0.403|0.488|0.879|0.681|0.660|0.933|0.411|0.866|0.910|
> |+ST (single head)|0.777|0.304|0.392|0.513|0.872|0.704|0.666|0.887|0.39|0.85|0.888|
>
> We found that using a single shared head is consistent with using separate heads in terms of performance boosts in function-centric tasks and varying effects on other tasks. Despite this result, we believe that investigating methods to improve cross-scale consistency is an important future direction, and have included a discussion of this in our future work section. We also include these results in Appendix A.3.1.
>
>
> > Q4: Can the authors provide examples where substructure-tuned embeddings correlate with known functional motifs or mechanistic insight?
>
> We thank the reviewer for raising this point, as the interpretability that comes from incorporating substructures with known functions is a key motivation for our work. **To understand correlation of substructure-tuned embeddings with known functional motifs, we performed an explainability analysis**:
>
> - We identified all proteins in the GO:MF test set whose GO:MF annotations could be attributed to a specific domain (accomplished via InterPro2GO [1] mappings), yielding 81 (protein, domain, GO:MF) triplets
> - For each protein, we computed a saliency map to calculate residue contributions to the prediction of the associated GO:MF term
> - We then calculated the total contribution of residues contained within the associated domain and normalized this by the total contribution across all residues in the protein
> - We performed this analysis for ESM-C 300M before and after substructure-tuning
>
> We found that **attributions to domain residues increased by an average of 17%**, indicating increased usage of substructures for function prediction after substructure-tuning. We believe this provides insight into the potential of substructure-aware representations for interpretability, and thank the reviewer for prompting us to pursue this analysis. We include this analysis, associated plots, and an illustrative visual example of increased attribution to domain residues in Appendix A.3.3.
>
> ---
>
> We greatly appreciate the reviewer and their feedback for helping us increase the depth and rigor of our work. We are happy to engage in further discussion, so please let us know if you have any additional feedback or points of concern with our work that would prevent you from increasing your score assessment.
>
> **References**: [1] https://www.ebi.ac.uk/GOA/InterPro2GO

---

> > ### Author Response · Authors · 2025-11-26
> > **Follow-up on response to reviewer w8Xw**
> >
> > Hello, we wanted to follow-up since the end of the discussion period is coming up in a week and we're also aware that a long holiday weekend is about to begin in the US. First, thank you for the time you've already spent engaging with our submission.
> >
> > We're very excited about the opportunity to share our work with the broader community, so we would be grateful if you're able to review our response to your feedback and share any additional questions or comments. If our response sufficiently addresses your concerns, we would also be thankful if you would consider updating your assessment accordingly. We understand that this is a large time commitment, so thank you so much for your time!

---

> ### Author Response · Authors · 2025-11-21
> **Response to reviewer w8Xw (1/2)**
>
> We thank the reviewer for their careful review and detailed feedback. We appreciate their assessment that our work **“identifies an under-explored gap”** in modern protein models, **“offers a novel, systematic way to integrate decades of curated biological knowledge into modern representation learning”**, and **“is both scientifically meaningful and practically useful, likely to influence future developments in protein representation learning, function prediction, and design.”**
>
> > W2: The dataset filters out substructures with < 75 occurrences.
>
> We agree that the choice to filter out rare substructures was a limitation of the work, and thus have explored relaxing the count cutoff. We found that **substructure-tuning is robust to expanding the dataset to rarer substructure types** and thus appreciate the reviewer’s feedback for prompting us to **expand the breadth of substructures represented** in our dataset. For the analysis and results, please refer to point P1 in the global response.
>
> > W3: are low-level features being overwritten, or are substructure objectives misaligned with certain downstream signals?
>
> We thank the reviewer for raising this key point, as understanding the trade-offs introduced by substructure-tuning increases the depth of our work. To explore this question, we performed a gradient conflict analysis, finding that **there is no misalignment with substructure-tuning and specific downstream evaluation tasks**. While this doesn’t provide "smoking-gun" evidence of why substructure-tuning hurts some tasks, we believe this provides a deeper understanding of substructure-tuning. For more details, please refer to point P3 in the global response.
>
> > W4 : The paper could better contextualize gains against concurrent structure-aware fine-tuning strategies (e.g., S-PLM, ESM-S).
>
> We thank the reviewer for raising this point and agree that a direct comparison of substructure-tuning to methods that use global structure to tune sequence models would help contextualize our work. To ensure a fair comparison, we have integrated ESM-S into Magneton and evaluated it on our benchmark suite:
>
> |Model|EC ($F_\max$)|GO:BP ($F_\max$)|GO:CC ($F_\max$)|GO:MF ($F_\max$)|Binary Localization (Accuracy)|Subcellular Localization (Accuracy)|Thermostability (Spearman's rho)|Human PPI (AUROC)|FLIP_bind ($F_\max$)|Biolip binding (AUROC)|Biolip catalytic (AUROC)|
> |---|---|---|---|---|---|---|---|---|---|---|---|
> |ESM2-150M|0.727|0.316|0.416|0.441|0.869|0.694|0.627|0.933|0.379|0.871|0.910|
> |+ST|0.742|0.324|0.415|0.473|0.866|0.679|0.582|0.919|0.327|0.852|0.890|
> |ESM-S 150M|0.760|0.315|0.373|0.460|0.866|0.677|0.631|0.917|0.352|0.871|0.885|
> |ESM2-650M|0.755|0.319|0.431|0.486|0.876|0.710|0.643|0.939|0.366|0.849|0.912|
> |+ST|0.745|0.321|0.440|0.534|0.895|0.749|0.655|0.935|0.362|0.851|0.927|
> |ESM-S 650M|0.794|0.327|0.422|0.497|0.896|0.679|0.653|0.912|0.363|0.876|0.902|
>
> We see that **substructure-tuning compares favorably to ESM-S**, with generally larger improvements on function-related tasks and smaller performance drop-offs in other tasks. We also note that **substructure-tuning is also useful for models that already incorporate global structure information**, something which has not been shown for ESM-S’s methodology.
>
> We have also integrated S-PLM, but experiments are still running due to the dated architecture of that work (lack of Flash Attention). We will post an additional comment here with these results once they’re available.
>
> Again, **we thank the reviewer for raising this point to help contextualize our work within the broader literature**, and include these comparisons in Appendix A.4.2 for future readers.
>
> > W5: While performance tables are comprehensive, confidence intervals or variance over multiple runs are missing
>
> This is an omission on our part and we appreciate the reviewer bringing this up to help raise the rigor of our work. Due to the time constraints of the rebuttal period, we haven’t yet updated all of the tables in our paper with uncertainty quantification, but commit to doing so in the camera-ready version of the manuscript. To give a sense of what to expect, **we calculated confidence intervals based on 100 rounds of bootstrapping for the evaluation metrics** for a representative run of substructure-tuning on ESM-C 300M:
>
> |Model|EC ($F_\max$)|GO:BP ($F_\max$)|GO:CC ($F_\max$)|GO:MF ($F_\max$)|Binary Localization (Accuracy)|Subcellular Localization (Accuracy)|Thermostability (Spearman's rho)|Human PPI (AUROC)|FLIP_bind ($F_\max$)|Biolip binding (AUROC)|Biolip catalytic (AUROC)|
> |---|---|---|---|---|---|---|---|---|---|---|---|
> |ESM-C + ST|0.794 $\pm$ 0.007|0.327 $\pm$ 0.003|0.402 $\pm$ 0.003|0.515 $\pm$ 0.007|0.891 $\pm$ 0.008|0.730 $\pm$ 0.006|0.640 $\pm$ 0.014|0.932 $\pm$ 0.016|0.357 $\pm$ 0.004|0.837 $\pm$ 0.001|0.900 $\pm$ 0.002|
>
> We believe this gives confidence that **the performance differences from substructure-tuning are due to meaningful model differences rather than random fluctuations.**

---

> > ### Author Response · Authors · 2025-12-01
> > **Follow-up with additional results**
> >
> > While we understand that the reviewer can no longer reply to our comments, we wanted to follow-up with the results of running S-PLM on our benchmark suite:
> >
> > |Model|EC ($F_\max$)|GO:BP ($F_\max$)|GO:CC ($F_\max$)|GO:MF ($F_\max$)|Binary Localization (Accuracy)|Subcellular Localization (Accuracy)|Thermostability (Spearman's rho)|Human PPI (AUROC)|Binding residue ($F_\max$)|Functional site prediction (binding, AUROC)|Functional site prediction (catalytic, AUROC)|
> > |---|---|---|---|---|---|---|---|---|---|---|---|
> > |ESM2-150M|0.727|0.316|0.416|0.441|0.869|0.694|0.627|0.933|0.379|0.871|0.910|
> > |+ST|0.742|0.324|0.415|0.473|0.866|0.679|0.582|0.919|0.327|0.852|0.890|
> > |ESM-S 150M|0.760|0.315|0.373|0.460|0.866|0.677|0.631|0.917|0.352|0.871|0.885|
> > |ESM2-650M|0.755|0.319|0.431|0.486|0.876|0.710|0.643|0.939|0.366|0.849|0.912|
> > |+ST|0.745|0.321|0.440|0.534|0.895|0.749|0.655|0.935|0.362|0.851|0.927|
> > |ESM-S 650M|0.794|0.327|0.422|0.497|0.896|0.679|0.653|0.912|0.363|0.876|0.902|
> > | S-PLM (650M) | 0.566 | 0.284 | 0.375 | 0.482 | 0.91 | 0.622 | 0.682 | 0.911 | 0.358 | 0.5 | 0.5 |
> >
> > We can see that **substructure-tuning compares favorably to S-PLM as well as ESM-S**. We thank the reviewer for flagging this point to improve the contextualization of our work, and add these results to Appendix Table A.12.

---

### Official Review · Reviewer_RdsB · 2025-10-31

**Soundness:** 4
**Presentation:** 3
**Contribution:** 3
**Rating:** 4
**Confidence:** 4

**Summary:**

Protein models are typically supervised with individual residues or full structures. However, there are various labeled substructures of active sites, binding sites, etc. that have not been previously used to train models. This paper introduces a dataset and framework to train sequence or structure-sequence models on these datasets. They show that the fine-tuned models improve performance.

**Strengths:**

Substructures are intuitive abstractions that captures various levels of protein modeling that are not immediately apparent from either structure or sequence.
The dataset developed by processing Swissprot is large and diverse, and is often overlooked in training protein models.

**Weaknesses:**

While there is some analysis of the dataset in Table 1, it is insufficient for a paper which proposes a dataset.
The results are impressive but raises concerns as to how much of the test set's labels are related or derived from the SwissProt annotations.
The authors finetune the model using elastic weight consolidation (EWC). However, it is not clear empirically how much EWC affects downstream performance.

**Questions:**

1. Beyond Table 1, what does the dataset consist of? What do the unique types consist of?
2. How do the authors address potential data leakage between the training data present in the Magneton substructure dataset and the evaluation datasets? Are there sequence similarity cutoffs to filter the training set to ensure that the training and testing proteins are sufficiently different?

---

> ### Author Response · Authors · 2025-11-21
> **Response to reviewer RdsB (1/2)**
>
> We thank the reviewer for the valuable feedback and for pointing out areas where our presentation was either unclear or incomplete. We also appreciate their assessment that our work pushes for training protein models on substructures “that are not immediately apparent from either sequence or structure” and “is often overlooked in training protein models”.
>
> > Q1: Beyond Table 1, what does the dataset consist of? What do the unique types consist of?
>
> We acknowledge that the previous description of the dataset lacked a significant amount of detail regarding the exact structure of the dataset, the meaning of the different substructure types (e.g. the difference between “active site” and “conserved site”), and the precise steps taken to create the dataset. To resolve this, we have **added a new appendix section (Appendix A.1.1) that includes the following**:
> - Detailed processing steps for creating the dataset, including URLs for original data files, descriptions of individual processing steps, and paths to relevant scripts and documentation in our GitHub repository
> - Descriptions of each substructure category along with illustrative examples of some of the different unique types within each category, to give a sense of what each substructure category represents
> - Example entry from the dataset with associated description
>
> > Q2: How do the authors address potential data leakage between the training data present in the Magneton substructure dataset and the evaluation datasets? Are there sequence similarity cutoffs to filter the training set to ensure that the training and testing proteins are sufficiently different?
>
> We agree that data leakage is a serious concern here and we appreciate the reviewer flagging this as an area to strengthen our presentation.
>
> For the Magneton substructure dataset, following established approaches in recent structure-based protein models [1], **we created sequence-based splits of our dataset using the AFDB50 sequence-based clusters** [2]. These clusters consist of sequences that share at least 50% sequence identity and 90% sequence overlap, **ensuring that the train and test set do not share proteins with significant sequence similarity**.
>
> Regarding overlap between the Magneton train set and the test sets of the evaluation benchmarks, we had not considered this. It’s standard practice when training protein models to consider the dataset used for self-supervised learning as distinct from the downstream evaluation sets (whether this is good practice is another question). For example, ESM2, GearNet, SaProt, and ProSST all do not address the possibility of leakage between the datasets used to train their models and the test sets for downstream evaluation tasks. Presumably this is because the sequence or structure data is considered sufficiently far removed from the labels used in downstream evaluations. However, **we agree that substructural annotations are different from sequence or global structure in that they can be more directly tied to protein function (which is also exactly why we wanted to incorporate them into protein models!)**.
>
> To assess whether this data leakage is responsible for any of our results, **we constructed a more stringent dataset split** by identifying all proteins in the Magneton train set that were also contained in the test split of any evaluation benchmark, and then removing those proteins and all of their corresponding AFDB50 clusters from the Magneton train set. This removed 31,191 of the 423,885 proteins in the Magneton training split.
>
> We next performed substructure-tuning of ESM-C 300M using active, binding, and conserved site annotations with this stringent training split and evaluated the resulting model on our evaluation suite:
>
> |Model|EC ($F_\max$)|GO:BP ($F_\max$)|GO:CC ($F_\max$)|GO:MF ($F_\max$)|Binary Localization (Accuracy)|Subcellular Localization (Accuracy)|Thermostability (Spearman's rho)|Human PPI (AUROC)|FLIP_bind ($F_\max$)|Biolip binding (AUROC)|Biolip catalytic (AUROC)|
> |---|---|---|---|---|---|---|---|---|---|---|---|
> |ESM-C 300M|0.688|0.307|0.416|0.429|0.871|0.703|0.648|0.917|0.367|0.851|0.923|
> |+ST (original)|0.761|0.325|0.403|0.488|0.879|0.681|0.660|0.933|0.411|0.866|0.910|
> |+ST (stringent)|0.789|0.317|0.403|0.5|0.88|0.687|0.647|0.892|0.372|0.847|0.879|
>
>
> The table above shows that using the more stringent data split has a net neutral effect on substructure-tuning, with some performance metrics increasing (e.g. EC, GO:MF) and others decreasing (e.g. binding and catalytic site prediction). We believe **this shows that data leakage does not drive any of the results presented in our main manuscript.** We also believe **this was a great point of feedback and strengthens the rigor of our work**, so we include these results in Appendix A.1.3 and make the stringent dataset splits available for use in our codebase.

---

> ### Author Response · Authors · 2025-11-21
> **Response to reviewer RdsB (2/2)**
>
> > Q2 continued
>
> Since there exist mappings from Gene Ontology terms to InterPro entries, we also checked whether substructure-tuning’s improvement on GO MF prediction could be attributed to this mapping. We used the InterPro2GO mapping [3] to ask if the improvement in predicting a GO MF term after substructure-tuning correlated with the representation of its corresponding domains in the Magneton train set. We provide a table here, but please refer to Appendix A.1.3 for full plots.
>
> Per-domain $F_\max$ improvement after substructure-tuning:
>
> | |Mean|25th percentile|Median|75th percentile|
> |---|---|---|---|---|
> |Domains in train set|+17.82%|-2.36%|+5.11%|+18.66%|
> |Domains not in train set|+16.84|-3.40%|+4.88%|+23.35%|
>
> We see that the **performance improvement for a GO MF term does not depend on its corresponding domain’s representation in the Magneton train set, supporting the lack of data leakage**. Again, we thank the reviewer for raising this point and helping increase the rigor of our work.
>
>
> > Q3: The authors finetune the model using elastic weight consolidation (EWC). However, it is not clear empirically how much EWC affects downstream performance.
>
> This is a good point and an omission on our part. Our use of EWC was motivated by an increase in the masked language modeling loss when performing substructure-tuning, as shown below for substructure-tuning ESM-C 300M with domain annotations:
>
> | |Frozen base model|Full finetuning|EWC finetuning|
> |---|---|---|---|
> |Validation MLM loss|1.24|2.30|1.75|
>
>
> Based on these findings, we incorporated EWC into the training procedure and observed that training with EWC offered a good balance of maintaining improvement on function-related tasks while mitigating performance decreases on other tasks:
>
> |Model|Substructures used for tuning|EC ($F_\max$)|GO:BP ($F_\max$)|GO:CC ($F_\max$)|GO:MF ($F_\max$)|Binary Localization (Accuracy)|Subcellular Localization (Accuracy)|
> |---|---|---|---|---|---|---|---|
> |ESM-C (300M)|N/A|0.688|0.307|0.416|0.429|0.871|0.703|
> |EWC|Domain|0.776|0.307|0.403|0.501|0.811|0.640|
> |No EWC|Domain|0.831|0.311|0.386|0.531|0.802|0.643|
>
> We have added these results and discussion to Appendix A.2.1 with a corresponding reference in the main text.
>
> ---
>
> We thank the reviewer again for the detailed feedback, which we believe has resulted in a much more complete and rigorous manuscript. We hope that our responses answer all of your questions, and if so, we kindly ask you to consider raising your score. Please don’t hesitate to let us know if you have any additional feedback or points of concern with our work that would prevent you from increasing your score assessment, as we’re excited to engage in a productive dialogue to improve our work. Thank you again!
>
> **References**: [1] Su et al., “SaProt: Protein Language Modeling with Structure-aware Vocabulary”, ICLR 2024. [2] Barrio-Hernandez et al., “Clustering predicted structures at the scale of the known protein universe”, Nature 2023. [3] https://www.ebi.ac.uk/GOA/InterPro2GO

---

> > ### Author Response · Authors · 2025-11-26
> > **Follow-up on response to reviewer RdsB**
> >
> > Hello, we wanted to follow-up since the end of the discussion period is coming up in a week and we're also aware that a long holiday weekend is about to begin in the US. First, thank you for the time you've already spent engaging with our submission.
> >
> > We're very excited about the opportunity to share our work with the broader community, so we would be grateful if you're able to review our response to your feedback and share any additional questions or comments. If our response sufficiently addresses your concerns, we would also be thankful if you would consider updating your assessment accordingly. We understand that this is a large time commitment, so thank you so much for your time!

---

> > > ### Comment · Reviewer_RdsB · 2025-11-27
> > > **Thanks for the response**
> > >
> > > > new appendix section (Appendix A.1.1): Detailed processing steps for creating the dataset, Descriptions of each substructure category, Example entry
> > >
> > > Thanks for the dataset description, this is very helpful. However, it lacks analysis. For example, what is the distribution of substructure sizes (number of residues) and amino acid compositions?
> > >
> > > > The sequence or structure data is considered sufficiently far removed from the labels used in downstream evaluations. However, we agree that substructural annotations are different from sequence or global structure in that they can be more directly tied to protein function
> > >
> > > I agree with this statement
> > >
> > > > To assess whether this data leakage is responsible for any of our results, we constructed a more stringent dataset split by identifying all proteins in the Magneton train set that were also contained in the test split of any evaluation benchmark, and then removing those proteins and all of their corresponding AFDB50 clusters from the Magneton train set. This removed 31,191 of the 423,885 proteins in the Magneton training split.
> > >
> > >
> > > Thanks for acknowledging my concerns surrounding data leakage. Does your filtering criteria ensure that no training and testing protein sequences have more than 30% sequence similarity? What is the maximum sequence similarity between the splits?
> > >
> > >
> > > > EWC results
> > >
> > > Thank you for the ablation. If I understand correctly, it seems like on some tasks (EC, GO:BP, GO:MP, subcellular localization), No EWC performs better, while for other tasks (GO:CC, Binary Localization) EWC helps. It is important to note this in the manuscript.
> > >
> > > ---
> > >
> > > If the authors are able to incorporate some sort of dataset analysis and confirm that the more stringent dataset filters result in low sequence similarity splits, I am happy to raise my score.

---

> > > > ### Author Response · Authors · 2025-12-01
> > > > **Response to follow-up questions**
> > > >
> > > > Thank you for the additional feedback, we really appreciate you engaging with us to help improve the rigor and overall quality of our work! Please find our answers to your additional questions below.
> > > >
> > > > > Thanks for the dataset description, this is very helpful. However, it lacks analysis. For example, what is the distribution of substructure sizes (number of residues) and amino acid compositions?
> > > >
> > > > We’re glad the additional descriptions were helpful for understanding the dataset. While we note that our original Table 1 contains the median number of residues and fraction of protein covered per substructure type, we also agree that the added dataset description is largely descriptive without much deeper insight into the distribution of substructures contained within the dataset. To remedy this, **we have added a new appendix section (Appendix A.1.2) containing exploratory analyses of the substructures themselves**. For each substructure type, we have added plots of:
> > > > - Number of residues per substructure
> > > > - Number of contiguous segments per substructure
> > > > - Fraction of protein covered per substructure
> > > > - Amino acid composition (across all 20 amino acids and grouped by sidechain chemical properties)
> > > >
> > > > We hope this additional analysis provides more intuition about the dataset.
> > > >
> > > > > Does your filtering criteria ensure that no training and testing protein sequences have more than 30% sequence similarity? What is the maximum sequence similarity between the splits?
> > > >
> > > > The filtering criteria we described in our first response only ensured that training and testing protein sequences roughly shared less than 50% sequence similarity. The 50% there comes from the construction of the clusters we used (AFDB50, reference [2] above) and the “roughly” comes from the approximate nature of the MMseqs2 clustering.
> > > >
> > > > To make sure our results hold up under a maximum of 30% sequence similarity, **we created another, even more stringent, dataset split** as follows:
> > > > - We took all of the proteins in the test set of any evaluation benchmark and aligned them to the Magneton training set proteins with MMseqs2
> > > > - We removed from the Magneton train set any protein with greater than 30% sequence similarity and 80% overlap to any of the evaluation test set proteins
> > > >
> > > > This removed 131,440 of the 423,885 proteins in the Magneton training split. With the explicit filtering in this new split, we can confirm that no training and testing protein sequences have more than 30% sequence similarity.
> > > >
> > > > We performed substructure-tuning with this new data split on a ESM-C 300M model with active, binding, and conserved site annotations:
> > > > | Model | EC ($F_\max$) | GO:BP ($F_\max$) | GO:CC ($F_\max$) | GO:MF ($F_\max$) | Binary Localization (Accuracy) | Subcellular Localization (Accuracy) | Thermostability (Spearman's $\rho$) | Human PPI (AUROC) | Binding residue ($F_\max$) | Functional site prediction (binding, AUROC) | Functional site prediction (catalytic, AUROC) |
> > > > | ----- | ----- | ----- | ----- | ----- | ----- | ----- | ----- | ----- | ----- | ----- | ----- |
> > > > | ESM-C 300M | 0.688 | 0.307 | 0.416 | 0.429 | 0.871 | 0.703 | 0.648 | 0.917 | 0.367 | 0.851 | 0.923 |
> > > > | \+ST (original) | 0.761 | 0.325 | 0.403 | 0.488 | 0.879 | 0.681 | 0.660 | 0.933 | 0.411 | 0.866 | 0.910 |
> > > > | \+ST (stringent) | 0.789 | 0.317 | 0.403 | 0.5 | 0.88 | 0.687 | 0.647 | 0.892 | 0.372 | 0.847 | 0.879 |
> > > > | \+ST (sim \< 30%) | 0.777 | 0.317 | 0.385 | 0.507 | 0.83 | 0.686 | 0.645 | 0.889 | 0.375 | 0.849 | 0.912 |
> > > >
> > > > We again see an overall net neutral effect from this extremely stringent data split, which we believe further strengthens the case for a lack of data leakage. **We believe this represents a level of rigor beyond the norm for protein representation learning work, and thank the reviewer for encouraging us to test the limits of our method**. We have added a description of the methods, the full results, and plots of sequence similarity distributions before and after filtering to Appendix A.1.4 in the updated manuscript. We also add these new splits to our publicly available dataset for future use by the community.
> > > >
> > > > > Thank you for the ablation. If I understand correctly, it seems like on some tasks (EC, GO:BP, GO:MP, subcellular localization), No EWC performs better, while for other tasks (GO:CC, Binary Localization) EWC helps. It is important to note this in the manuscript.
> > > >
> > > > Yes, we agree with your understanding. We believe that using EWC overall represents a tradeoff when substructure-tuning: reducing the performance improvements in tasks where it helps but also reducing the performance degradations in tasks where it hurts. **We have updated the manuscript to more explicitly note this** (Section 4.2, “Substructure-tuning across models”).
> > > >
> > > > ---
> > > >
> > > > While we understand the reviewer can no longer respond to our replies, we hope this answers the remaining questions, and once again, thank the reviewer for engaging with us to help improve the quality of our work.

---

### Official Review · Reviewer_DsWe · 2025-11-01

**Soundness:** 3
**Presentation:** 3
**Contribution:** 2
**Rating:** 6
**Confidence:** 3

**Summary:**

This paper presents Magneton, a framework that brings protein substructure knowledge into representation learning in a systematic way. The authors build a large annotated dataset with over half a million proteins and around 1.7 million substructures from six categories. They propose a substructure-tuning method that fine-tunes pretrained protein language models on multi-scale substructure classification and test its impact on 13 downstream tasks at different levels, from residues to protein interactions. The results show consistent improvements on function-related tasks such as EC, GO-MF, and thermostability, while the effects on localization and residue-level predictions are neutral or slightly negative. Overall, the work shows that substructure information complements global structural signals and offers a reproducible benchmark for studying hierarchical protein representations.

**Strengths:**

The paper takes a fresh approach to protein representation by explicitly modeling mid-level substructures that connect sequence and overall structure. While earlier work has explored structural supervision, few have looked at substructure information at this scale or systematically tested its impact. The authors run solid and wide-ranging experiments across several model families, including ESM2, SaProt, Cambrian, and ProSST, covering six types of substructures and thirteen downstream tasks. The dataset is clearly described and easy to reproduce, and the figures and tables make the framework intuitive to follow.

Overall, the work is clearly written and well-executed. It provides a practical and scalable benchmark that shows how substructure-level fine-tuning can bring meaningful gains on function-related tasks, even for models that already use 3D information. The results highlight the value of adding explicit substructure knowledge to protein models and point toward new directions for improving interpretability and representation learning in protein science.

**Weaknesses:**

Lack of theoretical motivation:
The paper shows that substructure-tuning works, but it doesn’t really explain why. There’s no clear reasoning or framework to describe why it helps with function prediction but hurts residue-level performance. The discussion feels more like observation than real analysis.

Unbalanced and oversimplified supervision:
The six types of substructures are not only uneven in number—some, like catalytic or binding sites, are much rarer—but also differ in biological meaning. Many of these categories, such as catalytic sites, cannot be treated as simple binary labels because they involve diverse biochemical functions rather than just “active vs inactive.” Similarly, tasks like PPI prediction or contact prediction represent complex relational signals that go beyond binary classification. The paper filters and simplifies these cases, but it doesn’t assess how this affects the model’s ability to capture functional diversity or relational structure.

**Questions:**

Mechanistic validation:
Could the authors include attention-map or embedding-similarity analyses (for example, t-SNE or clustering of substructure embeddings) to check whether substructure-tuning actually makes the model group residues or motifs by function? Comparing pre- and post-tuning representations on catalytic or binding site residues would make the mechanism more convincing.


Generalization and transferability:
To show robustness, the authors could run few-shot or zero-shot transfer experiments — for instance, testing on unseen protein families or on new substructure types not used during tuning. Measuring how performance drops as family divergence increases would make the generalization claims stronger.


Functional specificity of catalytic sites:
Instead of treating all catalytic sites as one binary label, the authors could run an additional experiment focusing on specific catalytic functions (for example, hydrolases vs oxidoreductases). This would test whether substructure-tuning helps the model distinguish between different catalytic mechanisms rather than just recognizing “catalytic vs non-catalytic” regions.

---

> ### Author Response · Authors · 2025-11-21
> **Response to reviewer DsWe**
>
> We thank the reviewer for their thoughtful feedback. We are also grateful that the reviewer finds our work to be “a fresh approach to protein representation”, “clearly written and well-executed”, and agrees that incorporating substructure into protein models “point toward new directions for improving interpretability and representation learning in protein science.”
>
> > W1: The paper shows that substructure-tuning works, but it doesn’t really explain why.
>
> This is a great point and prompted us to perform a deeper dive into the mechanics behind task-dependent effects of substructure-tuning. We performed a gradient conflict analysis, finding that **there is no misalignment with substructure-tuning and specific downstream evaluation tasks**. While this doesn’t provide “smoking-gun” evidence of why substructure-tuning hurts some tasks, we believe this provides a deeper understanding of substructure-tuning. For results and description of the analysis, please refer to point P3 in the global response.
>
> > W2.1: The six types of substructures are not only uneven in number—some, like catalytic or binding sites, are much rarer—but also differ in biological meaning.
>
> We agree that the six types of substructures included form an unbalanced dataset (e.g. millions of examples of secondary structure, tens of thousands of examples of active sites) and balancing these is something to think about when substructure-tuning with multiple substructure types simultaneously.
>
> Our current models are substructure-tuned until convergence of validation loss, so models trained on single substructure types should be comparable. Due to the large number of possible substructure-tuning configurations, our current exploration focused on which types to include at all rather than exploring the weights applied to different substructure types, but we agree that this is a current limitation and potentially fruitful avenue for future exploration, so we have added this to our “Limitations and Future Work” section.
>
> > W2.2 & Q3: categories, such as catalytic sites, cannot be treated as simple binary labels because they involve diverse biochemical functions rather than just “active vs inactive.” … test whether substructure-tuning helps the model distinguish between different catalytic mechanisms rather than just recognizing “catalytic vs non-catalytic” regions.
>
> We appreciate the reviewer raising this point, as capturing the diverse biological function of different substructure types is one of our core motivations. There may be some confusion here, so we apologize for a lack of clarity in our presentation. The site annotations used for substructure-tuning (active, binding, and conserved) are not binary labels, but rather categorical labels that distinguish between the diverse biochemical functions these sites are involved in. For example, [IPR001579](https://www.ebi.ac.uk/interpro/entry/InterPro/IPR001579/) corresponds to a glycosyl hydrolase active site and [IPR008255](https://www.ebi.ac.uk/interpro/entry/InterPro/IPR008255/) corresponds to a particular oxidoreductase active site, each of which are separate labels during substructure-tuning. The binary labels for binding and catalytic site residues come from a separate, independent dataset [1] that we incorporate into Magneton only for evaluating models and not for substructure-tuning. The **results in Table 3 show each model’s ability to distinguish between the different types of sites within a category in the multiclass setting**. However, we do agree that there is a gap here (and in the field) for a benchmark that probes a model’s ability to capture residue-level fine-grained functional information. **We have updated our methods section to make this distinction more clear**.
>
> > Q1 & Q2: check whether substructure-tuning actually makes the model group residues or motifs by function? …  the authors could run few-shot or zero-shot transfer experiments
>
> To investigate few- or zero-shot transfer, we calculated substructure embeddings for substructure types included during substructure-tuning as well as substructure types that were never seen during substructure-tuning. We found that  **substructure-tuning does make the model group residues or motifs by function** and that this finding is **also true for substructure types not seen during training**, indicating that **substructure-tuning encourages the models to learn general features of functional substructures, beyond just signatures of specific substructure types.** Please refer to point P2 in the global response for details of the analysis and results.
>
> ---
>
> We thank the reviewer for their careful reading of our manuscript and feedback that helped increase the depth of our work. We are happy to answer any additional questions and please don’t hesitate to share any concerns that might prevent you from increasing your score. Thank you again!
>
> **References:** [1] Yuan et al., “Protein Structure Tokenization: Benchmarking and New Recipe”, ICML 2025.

---

> > ### Author Response · Authors · 2025-11-26
> > **Follow-up on response to reviewer DsWe**
> >
> > Hello, we wanted to follow-up since the end of the discussion period is coming up in a week and we're also aware that a long holiday weekend is about to begin in the US. First, thank you for the time you've already spent engaging with our submission.
> >
> > We're very excited about the opportunity to share our work with the broader community, so we would be grateful if you're able to review our response to your feedback and share any additional questions or comments. If our response sufficiently addresses your concerns, we would also be thankful if you would consider updating your assessment accordingly. We understand that this is a large time commitment, so thank you so much for your time!

---

### Official Review · Reviewer_MQMA · 2025-11-02

**Soundness:** 3
**Presentation:** 3
**Contribution:** 3
**Rating:** 6
**Confidence:** 3

**Summary:**

The paper introduces Magneton, a new environment that incorporates protein substructures into protein language models. It contains (1) a large-scale dataset annotated with million-scale substructure annotations, (2) a suite of evaluation tasks,  and (3) a training method called substructure-tuning, a supervised fine-tuning method for distilling substructural information into pretrained models. Substructure-tuning shows substantial improvement on models’ ability to represent protein function.

**Strengths:**

1. Compiling curated substructure annotations at scale and providing packaging code + benchmark tasks is a valuable community resource.

2. The suite of 13 tasks spans from residue, substructure, protein, and interaction levels.

3. The paper reports consistent improvements on function-related tasks after substructure tuning, suggesting substructural priors are useful.

**Weaknesses:**

1. The authors restrict to substructures that occur at least 75 times in the SwissProt dataset, corresponding to retaining only the top 10% most frequently occurring domains. Although the histogram in Appendix A supports this, more experiments on this may be beneficial. Also, this practice might limit claims about the general utility of rare/novel substructures.

2. Have you tried other structure pooling techniques more than the mean pool?  E.g. learnable aggregation, or attention?

**Questions:**

Please refer to the weakness section.

---

> ### Author Response · Authors · 2025-11-21
> **Response to reviewer MQMA**
>
> We thank the reviewer for their feedback and the time they’ve spent helping us improve our work. We’re appreciative of their overall positive response to our work and are especially gratified by their assessment that our work presents “a valuable community resource”.
>
> In response to the specific suggestions provided, we’ve attempted to address the reviewer’s questions via additional experiments which show that our method is robust to inclusion of infrequent substructures and the pooling method used in substructure-tuning, as well as generalizes to rare or unseen substructures.
>
> > Q1: The authors restrict to substructures that occur at least 75 times in the SwissProt dataset, corresponding to retaining only the top 10% most frequently occurring domains. Although the histogram in Appendix A supports this, more experiments on this may be beneficial. Also, this practice might limit claims about the general utility of rare/novel substructures.
>
> The points about exploring the frequency cutoff used in our dataset is a great one and prompted us to find both that **substructure-tuning is robust to using rare substructures** and that by relaxing the cutoff we are able to **greatly increase the number of unique substructure types captured by our dataset**. For detailed results, please see point P1 in the global response.
>
> Regarding utility for rare/novel substructures, this was also a great point that spurred us to more deeply investigate the effects of substructure-tuning. We found that **substructure-tuning encourages models to learn general features of functional substructures, beyond just signatures of specific substructure types** and that these features **generalize to rare/novel substructures**. For a description of results and analysis, please see point P2 in the global response.
>
>
> > Q2: Have you tried other structure pooling techniques more than the mean pool? E.g. learnable aggregation, or attention?
>
> While we initially developed our approach using mean pooling to construct substructure representations due to its simplicity, we agree that exploration of additional aggregation techniques helps expand the completeness of our work. We note that the choice of a simple pooling over a learnable aggregation was intentional: since the goal of substructure-tuning is to use the supervised substructure classification problem to tune the parameters of the underlying protein model, we wanted to encourage learning to occur within the base model rather than in a parameterized pooling module.
>
> However, we agree that alternate aggregations could allow for more targeted updating of the base protein encoder than mean pooling, which necessarily coarsens residue-level information. To investigate this, we performed substructure-tuning using max pooling and a learnable attention pooling. Both of these experiments performed substructure-tuning on ESM-C 300M using active, binding, and conserved site annotations. The results are presented below as well as in Appendix A.4.1:
> |Model|EC ($F_\max$)|GO:BP ($F_\max$)|GO:CC ($F_\max$)|GO:MF ($F_\max$)|Binary Localization (Accuracy)|Subcellular Localization (Accuracy)|Thermostability (Spearman's $\rho$)|HumanPPI (AUROC)|
> |---|---|---|---|---|---|---|---|---|
> |ESM-C 300M|0.688|0.307|0.416|0.429|0.871|0.703|0.648|0.917|
> |+ST (original,mean)|0.761|0.325|0.403|0.488|0.879|0.681|0.660|0.933|
> |+ST (max)|0.777|0.304|0.392|0.513|0.872|0.704|0.666|0.887|
> |+ST (attention)|0.778|0.317|0.392|0.505|0.815|0.675|0.643|0.890|
>
>
> The table above shows that the **substructure-tuning is robust to the exact method used for pooling residue-level embeddings to substructural embeddings**. To allow other users to experiment with these settings, **we have introduced the choice of substructure pooling method as a configurable option in our codebase**.
>
> ---
>
> Again, we thank the reviewer for the helpful feedback and are happy to answer any additional questions or comments. We hope that our responses answer all of your questions, and if so, we kindly ask you to consider raising your score. Thank you again!

---

> > ### Author Response · Authors · 2025-11-26
> > **Follow-up on response to reviewer MQMA**
> >
> > Hello, we wanted to follow-up since the end of the discussion period is coming up in a week and we're also aware that a long holiday weekend is about to begin in the US. First, thank you for the time you've already spent engaging with our submission.
> >
> > We're very excited about the opportunity to share our work with the broader community, so we would be grateful if you're able to review our response to your feedback and share any additional questions or comments. If our response sufficiently addresses your concerns, we would also be thankful if you would consider updating your assessment accordingly. We understand that this is a large time commitment, so thank you so much for your time!

---

### Author Response · Authors · 2025-11-21
**Global response (2/2)**

### P3: **Mechanistic understanding of task-dependent effects of substructure-tuning** (DsWe, x8Xw)

To investigate the trade-offs introduced by substructure-tuning, particularly the improvement in function prediction but degradation in residue-level tasks, **we performed a gradient conflict analysis**. Using the ESM-C 300M base model, we computed gradients for the substructure classification task and for a set of evaluation tasks, including protein-level function prediction and residue-level classification. We measured cosine similarity between gradients across batches within each task and across different tasks. While gradients for all evaluation tasks were highly consistent across batches, gradients for substructure classification had lower, although still positive, within-task similarity and were close to orthogonal to gradients for the evaluation tasks.

|GO:MF|EC|Binding residue|Functional site prediction (binding)|Functional site prediction (catalytic)|Substructure classification|
|---|---|---|---|---|---|
|0.957 ± 0.004|0.956 ± 0.004|0.961 ± 0.007|0.964 ± 0.004|0.973 ± 0.022|0.095 ± 0.077|
Within task gradient similarity (mean $\pm$ std deviation)

|GO:MF|EC|Binding residue|Functional site prediction (binding)|Functional site prediction (catalytic)|
|---|---|---|---|---|
|-0.006 ± 0.047|0.003 ± 0.027|0.022 ± 0.056|0.002 ± 0.055|-0.005 ± 0.038|
Gradient similarity of evaluation task with substructure classification gradient (mean $\pm$ std deviation)

These results do not fully explain the task-specific effects of substructure-tuning, but they **suggest that the behavior is not due to a simple global misalignment between the substructure objective and certain downstream tasks.** Instead, we hypothesize that our current instantiation of substructure-tuning biases the model against fine-grained residue-level distinctions, because it explicitly encourages residues within the same substructure to share similar representations. As discussed in our Future Work section, this motivates the development of methods that incorporate substructure information at the architectural level or training objectives that operate jointly across spatial and functional scales. **We thank the reviewers for raising this point, which resulted in a better understanding of the mechanisms underlying substructure-tuning.** We provide a discussion of this analysis at the end of Section 4.2 and full details in Appendix A.3.2.

## Additional experiments
In addition to the experiments described above, we also conducted 6 additional experiments to investigate comments from individual reviewers:
1. Alternate pooling methods, including max pooling and a learnable attention aggregation. We found that substructure-tuning is robust to the choice of pooling method.
2. Verification of lack of data leakage.
    - We constructed new stringent dataset split that remove from training all proteins similar to those in the test set of any evaluation benchmark. This results in no change in substructure-tuning effects, indicating the lack of data leakage.
    - We also confirmed that improvements in GO MF prediction were not driven by representation of associated domains in the substructure training set.
3. Direct comparison against methods for tuning sequence-based models with global structure information (ESM-S and S-PLM). Substructure-tuning compares favorably to ESM-S. Results for S-PLM are pending and will be added as soon as available.
4. Exploration of a single, shared classification head across substructure types rather than separate heads per type. This configuration is consistent with separate heads.
5. An explainability analysis identifying correlations between substructure-tuned embeddings and known functional motifs. In GO:MF classification, substructure-tuning increases attribution to residues within associated domains by 17% on average.

We have also improved the overall presentation by adding additional context to our work:
1. Appendix section with detailed descriptions of Magneton dataset processing and contents
2. Motivation and experiments supporting inclusion of elastic weight consolidation
3. Bootstrapped uncertainty estimates for performance metrics

## Limitations discussed

We have also transparently discussed remaining limitations including: (1) architectural constraints of the fine-tuning approach, (2) dataset imbalance across substructure types, and (3) trade-offs between protein-level and residue-level performance. These points are now addressed in expanded "Limitations and Future Work" sections.

---

We believe these substantial additions, comprising 9 new experimental analyses, have significantly strengthened the scientific rigor, mechanistic understanding, and practical value of our work. We are committed to incorporating all feedback into the camera-ready version and look forward to further discussion.

---

### Author Response · Authors · 2025-11-21
**Global response (1/2)**

We genuinely appreciate the careful reading of our manuscript and detailed feedback from all reviewers.

We are encouraged by the reviewers’ positive assessment of our work as **“a valuable community resource"** (MQMA), a **“point toward new directions for improving interpretability and representation learning in protein science”** (DsWe), incorporating data that **“is often overlooked in training protein models”** (RdsB), and as **“both scientifically meaningful and practically useful, likely to influence future developments in protein representation learning, function prediction, and design”** (w8Xw). The feedback has helped improve the depth, rigor, and overall value of the work.

We provide a shared response here with experiments that are applicable to points raised by multiple reviewers. We itemize these points as "P{number}" and refer to these within the individual reviewer replies. We have also updated the PDF of our submission with all new additions indicated with blue text.

## Common points of feedback

### P1: **Consequences of restricting dataset to substructures that occur at least 75 times in SwissProt dataset** (MQMA, x8Xw)

This was a very helpful suggestion, as relaxing the count cutoff used for our substructure dataset presents an opportunity for expanding the breadth of substructures represented. **We generated new versions of our dataset using more permissive cutoffs (minimum counts of 25 or 10)**, performed substructure-tuning with these datasets, and evaluated the resulting models on our benchmark suite. This experiment used ESM-C 300M with substructure-tuning using active, binding, and conserved site annotations:

|Model|EC ($F_\max$)|GO:BP ($F_\max$)|GO:CC ($F_\max$)|GO:MF ($F_\max$)|Binary Localization (Accuracy)|Subcellular Localization (Accuracy)|Thermostability (Spearman's $\rho$)|Human PPI (AUROC)|
|---|---|---|---|---|---|---|---|---|
|ESM-C 300M|0.688|0.307|0.416|0.429|0.871|0.703|0.648|0.917|
|+ST (original,>=75)|0.761|0.325|0.403|0.488|0.879|0.681|0.66|0.933|
|+ST (>=25)|0.802|0.32|0.399|0.526|0.851|0.693|0.64|0.917|
|+ST (>=10)|0.792|0.327|0.401|0.514|0.89|0.728|0.64|0.852|


The table shows that the **performance of substructure-tuning is consistent across the different cutoffs used to construct the dataset. Reducing the cutoff down to 10 also greatly expands the number of substructure types included in the dataset:**

|Substructure type|Full SwissProt set|Count >=75|Count >=25|Count >=10|
|---|---|---|---|---|
|Homologous superfamily|3511|1133|1685|2159|
|Domain|15868|917|1964|3506|
|Conserved site|748|356|573|691|
|Binding site|76|48|72|74|
|Active site|133|82|114|127|

These results are presented in Appendix A.1.2 and we have **added these expanded label sets to our dataset for public use** and believe the increased number of substructure types will be appreciated by the community.

### P2: **Generalization to rare or novel substructures** (MQMA, DsWe, x8Xw)

The question of whether substructure-tuning generalizes to unseen substructures is a great one. To explore this, we used our original dataset split to construct two sets of substructure types:
- “Seen” substructure types that were included during substructure-tuning (>= 75 occurrences in the SwissProt dataset)
- “Unseen” substructure types that were excluded during substructure-tuning, and further filtered to remove extremely low sample size types (< 75 and >= 10 occurrences in the SwissProt dataset)

We computed embeddings for all examples of these substructure types in the Magneton test set, before and after substructure-tuning, and calculated silhouette scores to measure clustering of the embeddings of each substructure type (higher score indicates tighter grouping of substructures of the same type). We used ESM-C 300M and SaProt as representative sequence and sequence-structure models for this analysis.

Average silhouette score across classes:
|Model|Homologous superfamily||Domain||Conserved site||Binding site||Active site||
|---|---|---|---|---|---|---|---|---|---|---|
| |Seen|Unseen|Seen|Unseen|Seen|Unseen|Seen|Unseen|Seen|Unseen|
|ESM-C|-0.183|0.180|-0.184|0.201|0.279|0.466|0.378|0.641|0.490|0.476|
|+ ST|0.339|0.584|0.486|0.652|0.830|0.747|0.882|0.894|0.933|0.816|
|SaProt|0.079|0.301|0.122|0.412|0.534|0.623|0.613|0.796|0.714|0.701|
|+ ST|0.478|0.684|0.554|0.717|0.796|0.764|0.843|0.938|0.912|0.866|

**Substructure-tuning results in more consistent representations of both seen and unseen substructures, despite never training on any examples of the unseen substructure types**. These results indicate that **substructure-tuning encourages the models to learn general features of functional substructures, rather than just signatures of specific substructure types**. In some cases, unseen substructures cluster more tightly than seen substructures, which we hypothesize is due to the distinctiveness of these rarer substructure types. We include these results in the main text at the end of Section 4.2.

---

### Author Response · Authors · 2025-12-01
**Table header clarification**

Apologies for any confusion, but we have noticed that some of the tables we shared in our replies used the names for some of the benchmark tasks as they appear in our codebase rather than as they're displayed in our manuscript. To clarify:

- "FLIP_bind" corresponds to the "Binding residue" task in Table 6 of our manuscript
- "Biolip binding" and "Biolip catalytic" correspond to the "Functional Site Prediction [Binding|Catalytic]" tasks in Table 6 of our manuscript

We hope this helps clarify the results shared in our replies.

---

### Author Response · Authors · 2025-12-04
**Summary of discussion phase for AC (1/2)**

Here we provide a summary of the reviews and our responses for the area chair. First, we thank the reviewers for the thoughtful feedback which helped greatly improve the depth and rigor of our work. Second, we thank the AC for their time and effort reviewing our work, especially in light of the increased workload this year.

## Summary of our work
Our work poses the open question: *how should we systematically incorporate decades of biological knowledge about protein substructures into protein encoding models?* Proteins are composed of recurrent, evolutionarily conserved, substructures that mediate core molecular functions, but the majority of recent protein modeling work ignores this body of knowledge. To fill this gap, we introduce Magneton, an integrated environment for developing substructure-aware protein models. Magneton consists of a large-scale dataset of ~530k proteins with over 1.7M annotated substructures, a training framework for incorporating substructure information into existing models, and a benchmark suite of 13 tasks probing learned representations at the residue-, substructure-, and protein-level. Using Magneton, we develop substructure-tuning, a supervised fine-tuning method that distills substructural knowledge into pretrained protein models. Across state-of-the-art sequence- and structure-based models, substructure-tuning
improves function-related tasks while revealing that substructural signals are complementary to global structural information.

## Summary of strengths
Overall, reviewers found our work to be a fresh perspective on protein modeling that is both clearly presented as well as rigorously executed. A few quotes from reviewers to support this summary:
> The proposed Magneton environment and “substructure-tuning” paradigm offer a novel, systematic way to integrate decades of curated biological knowledge into modern representation learning (w8Xw)

> Overall, the work is clearly written and well-executed (DsWe)

> The results … point toward new directions for improving interpretability and representation learning in protein science (DsWe)

> a fresh approach to protein representation (DsWe)

> This contribution is both scientifically meaningful and practically useful, likely to influence future developments in protein representation learning, function prediction, and design. (w8Xw)

> a valuable community resource (MQMA)

> suggesting substructural priors are useful (MQMA)

> The dataset developed by processing Swissprot is large and diverse, and is often overlooked in training protein models. (RdsB)

We also note that reviewer RdsB commented “If the authors are able to incorporate some sort of dataset analysis and confirm that the more stringent dataset filters result in low sequence similarity splits, I am happy to raise my score” shortly before the reviewer replies were disabled due to deanonymization concerns. We believe our subsequent additions addressed both of these concerns.

---

> ### Author Response · Authors · 2025-12-04
> **Summary of discussion phase for AC (2/2)**
>
> ## Summary of additions
> In response to the thoughtful feedback from reviewers, we made the following additions and edits to our manuscript, comprising **3 expanded appendix sections for greater depth and 10 new experiments which resulted in one new main text table, 16 appendix figures, and 12 appendix tables**. In our revised PDF, all additions from the discussion phase are shown in blue text.
>
> ### Additional experiments:
> - Expansion of dataset: we expanded our substructure dataset to rarer substructure types, which both increased the number of substructure types represented in our dataset as well as demonstrated our method’s robustness to very infrequent substructure types
> - Generalization to rare and unseen substructures: we demonstrated that substructure-tuning increases a model’s ability to group substructures of the same type, for both substructure types seen during training as well as those completely held out from training. This demonstrates that substructure-tuning encourages the models to learn general features of functional substructures, rather than just signatures of specific substructure types. We show this for both sequence-only and sequence-structure models.
> - Mechanistic understanding of task-specific substructure-tuning effects: we performed a gradient conflict analysis to better understand why substructure-tuning helps some tasks and hurts others. We found that task-specific effects are not due to misalignment with substructure-tuning, which we believe motivates future development of methods that incorporate substructure information at the architectural level.
> - Lack of data leakage: since protein substructures can be thought to have a closer relationship to protein function than sequence or global structure alone, we construct two new dataset splits that ensure minimal overlap between the substructure training set and the test sets of our benchmark tasks. We demonstrate that our method’s improvements hold up even under these stringent evaluation settings.
>   - Additionally, we demonstrate that our improvements in GO:MF prediction cannot be trivially attributed to representation of associated domains in our substructure training set.
> - Improved interpretability from substructure-tuning: we perform an explainability analysis, finding that substructure-tuning increases feature attribution to amino acids contained within functional domains when performing GO:MF prediction.
> - Exploration of additional pooling mechanisms for substructure-tuning (learnable attention pooling, max pooling)
> - Comparisons to existing methods that tune sequence-based models with global structure information (ESM-S, S-PLM)
>
> ### Expanded content
> - Additional descriptions of dataset construction and contents
> - Motivation and experiments supporting inclusion of elastic weight consolidation
> - Bootstrapped uncertainty estimates for performance metrics (to be added in camera-ready version)
>
> ### Expanded discussion within manuscript
> - Future development of architectural modifications or novel training objectives to incorporate substructural information
> - Dataset imbalance across substructure types
> - Trade-offs between protein-level and residue-level performance
>
> ---
>
> Overall, we believe these extensive additions and modifications to our manuscript have resulted in a work that is both more rigorous and of greater value to the community. We appreciate the reviewers’ deep engagement with our work and the thoughtful feedback.

---

### Meta-Review · Area_Chair_FuwL · 2026-01-07

**Summary:**

The reviewers had concerns about leaving out rare substructures, generalization to unseen substructures, and a lack of mechanistic understanding.

**Reviewer Concerns:**

**Neglect of rarer substructures**

The rebuttal includes experiments that finetune on much rarer substructures, demonstrating the robustness of substructure finetuning.

**Generalization to rare or unseen substructures**

The rebuttal shows that embeddings for rare and unseen substructures are more consistent after substructure finetuning.

**Understanding why substructure finetuning helps some tasks but not others**

The rebuttal analyzes gradient consistency to show that there is not a fundamental incompatibility between substructure finetuning and tasks where it does not improve performance. It is still not necessarily clear why substructure finetuning does not always improve performance.

**Insufficient description of the dataset**

The authors add an appendix providing further analysis of the dataset.

Overall, this is a novel and principled dataset, the most important concerns were addressed, and then experiments are sufficient to show that the dataset can be useful.

**Reviewer Scores:**

w8Xw: 6 -> 8
RdsB: 4 -> 6
DsWe: 6 -> 8
MQMA: 6 -> 8

---

### Decision · Program_Chairs · 2026-01-26

Accept (Poster)